# The functional specialization of visual cortex emerges from training parallel pathways with self-supervised predictive learning

**Shahab Bakhtiari**
Mila & McGill University
bakhtias@mila.quebec

**Patrick Mineault**
patrick.mineault@gmail.com

**Tim Lillicrap**
DeepMind
timothylillicrap@google.com

**Christopher C. Pack**
McGill University
christopher.pack@mcgill.ca

**Blake A. Richards**
CIFAR, Mila & McGill University
blake.richards@mila.quebec

## Abstract

The visual system of mammals is comprised of parallel, hierarchical specialized pathways. Different pathways are specialized in so far as they use representations that are more suitable for supporting specific downstream behaviours. In particular, the clearest example is the specialization of the ventral ("what") and dorsal ("where") pathways of the visual cortex. These two pathways support behaviours related to visual recognition and movement, respectively. To-date, deep neural networks have mostly been used as models of the ventral, recognition pathway. However, it is unknown whether both pathways can be modelled with a single deep ANN. Here, we ask whether a single model with a single loss function can capture the properties of both the ventral and the dorsal pathways. We explore this question using data from mice, who like other mammals, have specialized pathways that appear to support recognition and movement behaviours. We show that when we train a deep neural network architecture with two parallel pathways using a self-supervised predictive loss function, we can outperform other models in fitting mouse visual cortex. Moreover, we can model both the dorsal and ventral pathways. These results demonstrate that a self-supervised predictive learning approach applied to parallel pathway architectures can account for some of the functional specialization seen in mammalian visual systems.

## 1 Introduction

In the mammalian visual cortex information is processed in a hierarchical manner using two specialized pathways [15, 52]: the ventral, or "where" pathway, and the dorsal, or "what" pathway. These two pathways are specialized for visual recognition and movement, respectively [42, 24, 19, 58, 59, 17]. For example, damage to the ventral pathway may impair object recognition, whereas damage to the dorsal pathway may impair motion perception [64].

Deep artificial neural networks (ANNs) trained in a supervised manner on object categorization have been successful at matching the representations of the ventral visual stream [62, 57, 32]. They have

been shown to develop representations that map onto the ventral hierarchy, and which can be used to predict [62] and control [1, 49] neural activity in the ventral pathway. However, when we look at the other principal visual pathway in the mammalian brain, i.e. the dorsal pathway, the situation is different. Very few studies have examined the ability of deep ANNs to develop representations that match the dorsal hierarchy (though see the following fMRI study: [21]). Moreover, to the best of our knowledge, no studies have demonstrated both ventral-like and dorsal-like representations in a single network.

This lack of deep ANN models that capture both ventral and dorsal pathways leads naturally to an important question: under what training conditions would specialized ventral-like and dorsal-like pathways emerge in a deep ANN? Would a second loss function be required to obtain matches to dorsal pathways, or is there a single loss function that could induce both types of representations?

One promising set of candidates are predictive self-supervised loss functions [45, 22, 38]. Recent work has shown that self-supervised learning can produce similar results to supervised learning for the ventral pathway [63, 34, 26]. Moreover, there is a large body of work showing that mammals possess predictive processing mechanisms in their cortex [31, 8, 30, 18, 16], including in the dorsal pathway [2, 35], which suggests that a predictive form of self-supervised learning could potentially lead to the emergence of both ventral and dorsal-like representations.

Addressing this question requires recordings from different ventral and dorsal visual areas in the brain. Here, we explore these issues using publicly available data from the Allen Brain Observatory [9], which provides recordings from a large number of areas in mouse visual cortex. We examine the ability of a self-supervised predictive loss function (contrastive predictive coding [45, 22]) to induce representations that match mouse visual cortex. When we train a network with a single pathway, we find that it possesses more ventral-like representations. However, when we train a network with two parallel pathways we find that the predictive loss function induces distinct representations that map onto the ventral/dorsal division. This allows the network to better support both object categorization and motion recognition downstream tasks via the respective specialized pathways. In contrast, supervised training with an action categorization task only leads to matches to the ventral pathway, and not the dorsal pathway.

Altogether, this work demonstrates that the two specialized pathways of visual cortex can be modelled with the same ANN if a self-supervised predictive loss function is applied to an architecture with parallel pathways. This suggests that self-supervised predictive loss functions may hold great promise for explaining the functional properties of mammalian visual cortex.

## 2    Background and related work

**Self-supervised ANN models of the ventral visual stream**    Recently, [63] and [34] showed that the representations learned by self-supervised models trained on static images produce good matches to the ventral pathway. Our work builds on this by exploring the potential for self-supervised learning to also explain the dorsal pathway.

**ANN models of the mouse visual system**    Mice have become a common animal model in visual neuroscience due to the sophisticated array of experimental tools [44, 27]. As such, [53] compared the responses of different areas in mouse visual cortex with the representations of a VGG16 trained on ImageNet. In this paper, we show that self-supervised learning can produce better fits to both ventral and dorsal areas than supervised learning.

**ANN models of the dorsal pathway**    The MotionNet ANN model [51], is a feedforward ANN trained to predict the motion direction of segments of natural images, which was inspired by the role of the dorsal pathway in motion perception. In [6], they show that learning both ventral and dorsal-like representations in a single ANN with two pathways is possible if one forces the two pathways to process the phase and amplitude of a complex decomposition of the stimuli separately. In [21], they show that training a deep ANN on supervised action recognition can induce some match to dorsal pathway fMRI recordings. In this study, we show that with prediction as the learning objective and an architecture that has two parallel pathways, both ventral-like and dorsal-like representations can be learned.

# 3  Methods

## 3.1  Datasets

We use the Allen Brain Observatory open 2-photon calcium imaging dataset for the experiments in this study. We select subsets of the dataset based on brain area, recording depth, and visual stimuli used. Recordings from five areas of mouse visual cortex are used (VISp, VISlm, VISal, VISpm, VISam; Figure 1a). We only exclude one area from our analyses (VISrl) because it is a multi-sensory area, and visual stimuli alone do not drive it well [9]. We use recordings from cortical depths of $175$-$250\mu m$ (which corresponds to cortical layers 2-3) as these are the recordings with the largest number of neurons. When selecting visual stimuli, we use recordings elicited by the presentation of natural movies, because unlike static images, movies can elicit clear responses in both ventral and dorsal areas [9]. Thus, we only use those parts of the dataset in which natural movies (30 seconds long) were presented as visual stimuli. More details about the dataset can be found in [9]. For training the deep ANNs, we use the UCF101 dataset [56] (see section A.4 of supplementary materials).

## 3.2  Analysis techniques

**Representational Similarity Analysis (RSA)**    To measure the representational similarities between real brains and ANNs we use the RSA method. RSA has been commonly used in both the neuroscience [11, 33] and deep learning literature [41, 43]. The details of RSA can be found in those citations, but we will summarize it briefly here. First, we create response matrices $R \in \mathbb{R}^{N \times M}$ for every brain area and every layer of our ANNs, where $N$ is the number of neurons and $M$ is the number of video blocks, with each block comprising 15 frames of the video. Element $ij$ of the response matrix represents the response of the $i^{th}$ neuron to the $j^{th}$ video sequence. We then use Pearson correlation to calculate the similarity of every pair of columns (*e.g.* $k^{th}$ and $l^{th}$ columns) in matrix $R$, and form the $M$ by $M$ Representation Similarity Matrix ($RSM \in \mathbb{R}^{M \times M}$), in which every element ($kl$) quantifies the similarity of the responses to the $k^{th}$ and $l^{th}$ videos blocks. Thus, the $RSM$s describe the representation space in a network, be it a brain area or an ANN layer. Given two $RSM$s, we then use Kendall's $\tau$ between the vectorized $RSM$s to quantify the similarity between the two representations. It is important to note that measurement noise can potentially induce bias in the RSM estimations, but since the variance of measurement noise is not expected to be different across different video blocks, using Kendall's $\tau$ rank correlation should cancel the bias in our RSM estimations (see [12] for more details). As an additional sanity check, we use RSA to compare the representations between mice. If RSA is identifying salient representational geometries, then the $RSM$s between different areas should be lower than the RSMs for the same areas. Indeed, as shown in Figure 1b, the representational similarity is highest for the same regions across animals (i.e. the diagonal values are larger). These diagonal values also represent the noise ceiling for these areas (see section A.1). Throughout the paper (with the exception of Figure 1b), $RSM$ similarities are reported as the percentage of the noise ceiling.

**Identification of brain regions in the hierarchy**    The hierarchical organization of mouse visual areas can be inferred based on the anatomical and functional properties of each brain region. We adopted an approximation of hierarchical indices as reported in [23]. The relative hierarchical placing of the visual areas included in this study are shown in Figure 1c.

**Identification of brain regions into ventral and dorsal streams**    We group the brain regions into two sets, i.e. ventral and dorsal areas. Although the ventral/dorsal specialization of visual pathways in mice is not as clear and well understood as it is in primates, many anatomical and physiological studies do suggest that mice also possess such specialized pathways [40, 58, 39, 37]. According to previous anatomical and physiological studies [17, 58], VISlm and VISam can be considered as the most ventral and dorsal areas of mouse visual cortex, respectively. We then use the similarity of representations between other areas and VISlm and VISam to estimate a ventral score (V-score = $r_{lm}^{A}/(r_{am}^{A} + r_{lm}^{A})$) and a dorsal score (D-score = $r_{am}^{A}/(r_{am}^{A} + r_{lm}^{A})$) for each area. $r_{am}^{A}$ is the representational similarity between area $A$ and VISam, and $r_{lm}^{A}$ is the representational similarity between area $A$ and VISlm. The D-score and V-score values for the five visual areas are plotted in Figure 1d. Based on the D-score and V-score values, we grouped VISlm, VISp, and VISpm as more ventral and VISal, and VISam as more dorsal in our analysis. This grouping is in keeping

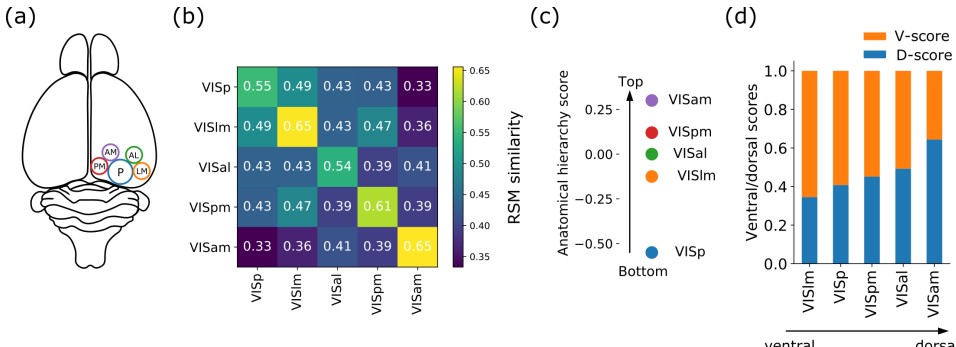

Figure 1: (a) The schematic of mouse visual cortex. (b) Representational similarity analysis between the visual areas included in our analysis. (c) The anatomical hierarchy score of the visual areas adopted from [23]. (d) Ventral and dorsal scores of the visual areas. Areas are sorted from the most ventral (VISlm - left) to the most dorsal (VISam - right) areas.

with a recent study that showed VISam and VISal are the first and second most responsive areas to motion stimuli: an important characteristic of dorsal areas [54].

## 3.3 Model

**Self-supervised predictive learning**  We used Contrastive Predictive Coding (CPC) for self-supervised learning, which was developed for modeling sequential data, including video datasets [45]. The loss function relies on predicting the future latent representations of a video sequence, given its present and past representations. See Figure S1 for a schematic of the model, and A.2 for more details regarding the CPC loss function.

**ANN Architecture**  All the ANN backbones used in our study are variants of the ResNet architecture, similar to the ones used in [13]. Our ResNet architectures have either one or two pathways. The one-pathway ResNet (ResNet-1p) is a regular 3D ResNet (Figure 2a). The two-pathway ResNet (ResNet-2p) is composed of two parallel ResNet branches, which split after a single convolutional layer, and merge after their final layers through concatenating their outputs along the channel dimension (Figure 3a). Both pathways of ResNet-2p receive a copy of the first convolutional layer output, and each has 10 Res-blocks. To keep the total number of output channels the same in both architectures, each pathway in ResNet-2p has half the number of channels of the single pathway in ResNet-1p. For the ANNs trained on object categorization, we use ResNet-18 pretrained on the ImageNet database [25]. Our 3D ResNet architectures are summarized in Table S1.

**Baseline models**  In terms of alignment with brain representations of video sequences, we compare the CPC trained ANNs with four other models: (1) a simple model based on Gabor filters, (2) a randomized deep ANN, (3) a ResNet-18 trained on ImageNet, and (4) 3D ResNets trained on action recognition in a supervised manner. See section A.3 of supplementary materials for more details.

**Training**  We use the backpropagation algorithm and Adam optimizer. CPC is trained with a batch size of 40 samples, a learning rate of $10^{-3}$, and 100 epochs. Supervised action recognition is trained with a batch size of 256 samples, a learning rate of $5 \times 10^{-4}$, and 300 epochs. All the models are implemented with PyTorch 1.7 and trained on RTX8000 NVIDIA GPUs.

**Downstream tasks**  In addition to comparisons with dorsal and ventral representations in mouse brain, we examine the two pathways of our trained ResNet-2p on two downstream tasks: object categorization and motion discrimination, which are supported in the real brain by the ventral and dorsal pathways, respectively. See section A.5 of supplementary materials for more details.

# 4 Results

## 4.1 CPC with a single pathway architecture produces better matches to mouse ventral stream

We first compare the representations learned with CPC on a single pathway architecture (Fig. 2a) with the three ventral areas of mouse visual cortex (VISlm, VISp, and VISpm; based on D-scores/V-scores in Figure 1d). In Figure 2b (top), the similarity of RSMs between different layers of the ANN and these three areas are shown. Comparing the maximum representational similarities between models in Figure 2c, we can see that, for the three ventral areas, CPC shows a higher maximum similarity compared to the other models. Compared to the baseline models (untrained ANN and Gabors), the ANN trained with object categorization has higher representational similarity to VISlm and VISp, the two most ventral areas (see Figure 1d), which is consistent with the suggested role of ventral areas in form and shape representation [10, 17]. This is in contrast with a previous study [5] that could not find any significant difference between ANNs trained on object categorization and untrained ANNs in modeling mouse visual cortex. There are several sources of variability that could explain this contradiction (e.g. architecture, datasets, etc.), but an important possibility is that the two studies used different visual stimuli, namely, natural videos in our study vs static natural images in [5]. Natural videos are better suited for eliciting sufficiently strong responses that can distinguish between the representations of the object categorization-trained and randomly initialized ANNs [9].

As noted, and as can be seen in Figure 2b, the maximum similarity happens in different hierarchical levels of the CPC-trained ANN for each area. We quantify the hierarchy index for every area by dividing the layer number with the highest similarity by the maximum number of layers. Based on this measure, the areas at the very top and bottom of the visual hierarchy have a hierarchy index of 1 and 0, respectively. The hierarchy index for the three areas are shown in Figure 2d. In accordance with the anatomical and functional data (Figure 1c), VISp has a lower hierarchical index ($0.41 \pm 0.15$) than VISlm ($0.61 \pm 0.08$) and VISpm ($0.55 \pm 0.08$). It should be noted, that despite the fact that anatomical/functional hierarchy indices position VISpm higher in the hierarchy than VISlm, our model-based measure of hierarchy places the two areas at around a similar level, with VISlm having a marginally higher value than VISpm. We speculate that this may be a result of the fact that VISpm has noisier representations, and therefore a lower noise ceiling (Fig. 1b). This may be due to VISpm actually being more multi-modal than purely visual.

## 4.2 CPC on a single pathway does not match mouse dorsal stream

We compare the similarity of RSMs between ResNet-1p trained with CPC (Fig. 2a) and the dorsal areas (VISam and VISal; based on D-scores/V-scores in Figure 1d). As shown in Figure 2b (bottom), the ANN trained with CPC just passes the pixel-level representations in its early layers (the first gray circle in the plots), and then the similarity values continuously decrease for deeper layers. Notably, for the most dorsal area (VISam), the maximum similarity of RSMs does not go above the untrained model (Figure 2c). For VISal, even though CPC shows some improvement compared to an untrained ANN, the performance is much lower than for the ventral areas, as shown in the previous section (Fig. 2c). These findings show that the representations learned by CPC and the ResNet-1p architecture are more ventral-like, and do not easily explain dorsal area representations (see supplementary section E). Similarly, an object categorization trained ANN has a very low similarity with the dorsal areas (even lower than the untrained ANN for VISam) indicating that object categorization cannot be considered as an appropriate loss function for the dorsal pathway, which is consistent with our current understanding in neuroscience.

## 4.3 An architecture with parallel pathways trained with CPC can model both ventral and dorsal areas

CPC's tendency to learn ventral-like representations, as seen in the previous section, could be due to a limitation of the backbone architecture. Predicting the next frame of a video with a contrastive loss requires learning invariances to every kind of transformation (or augmentation) that could happen from one frame to the next. For example, representing objects' shapes requires invariance to objects' motion, and representing objects' motion requires invariance to objects' shape. As suggested in [6], these two types of invariances often underlie the distinction between the dorsal and ventral representations in the brain. Therefore, one possibility is that ventral and dorsal-like representations

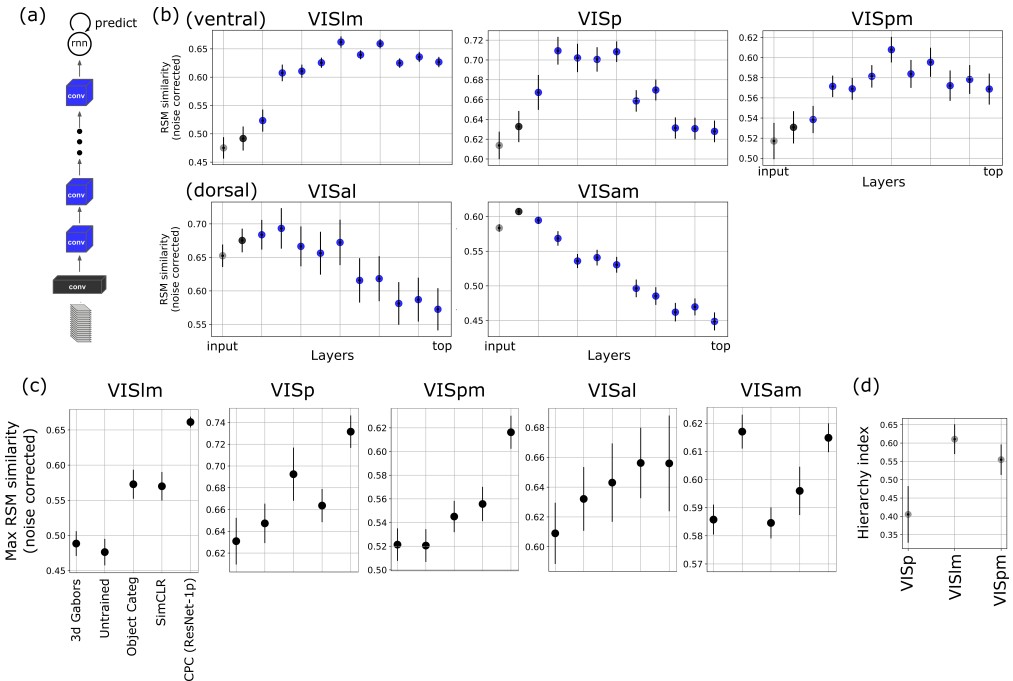

Figure 2: Representational Similarity Analysis between all the visual areas and the ANN trained with CPC. (a) The schematic of the ANN architecture with one pathway (ResNet-1p) used as the backbone of CPC. (b) Representational similarity between all the layers of the ANN with one pathways (trained with CPC) and the ventral (top: VISlm, VISp, VISpm) and the dorsal (bottom: VISal, VISam) areas. (c) The maximum representational similarity values between the ANN and the ventral and dorsal areas. (d) Hierarchy index of the ventral areas based on their fit to the ANN. Error bars represent bootstrapped standard deviation.

compete for resources in the network, and ultimately, the ventral-like representations win out, possibly because they provide a greater overall reduction in loss. If this hypothesis is true, then a network with two separate pathways may be able to reduce the competition by assigning one pathway to be more ventral-like, and one more dorsal-like.

To test this hypothesis, we use the simplest extension of ResNet-1p: ResNet-2p, a ResNet architecture that is composed of two identical, parallel ResNets which split after the first layer and merge after their final layers (Fig. 3a; see also section 3.3). Each pathway of ResNet-2p has half the number of channels of ResNet-1p, keeping the total number of channels per layer equal between the two architectures. We choose this ResNet-2p architecture as it shares all of the features of ResNet-1p, but with two separate pathways that could potentially be assigned specialized functions. We then check whether ResNet-2p can learn both ventral- and dorsal-like representations. We compare the representations of all the layers of the two pathways in ResNet-2p (blue and red in the schematic in Fig. 3a) with all the visual areas. The results show that the representations along one of the pathways (the blue pathway in Fig. 3a-b, top) are more ventral-like, and the representations along the other pathway (the red pathway in Fig. 3a-b, bottom) are more dorsal-like. Therefore, the two pathways together can model both ventral and dorsal areas of mouse visual cortex. Compared to an untrained ANN, both ResNet-1p and ResNet-2p achieve high RSM similarity values for ventral areas (Figure 3c), with ResNet-2p showing a slight decrease in VISlm and VISpm compared to ResNet-1p. However, unlike ResNet-1p, which fails to model the dorsal areas, ResNet-2p outperforms ResNet-1p and the untrained ANN for area VISam by a wide margin (Fig. 3c). For VISal, maximum RSM similarity values for ResNet-1p and ResNet-2p are around the same level (Fig. 3c), though the similarity values of the red pathway representations do not drop as much throughout the network as the blue pathway representations do, indicating the general similarity of the red pathway to the dorsal areas. Examination of the RSM similarity between the two pathways shows that the representational geometries are quite different (Fig. S2). Moreover, when we compare the RSMs of the two pathways with those from ResNet-1p, we can see that ResNet-1p has representations that are a better match

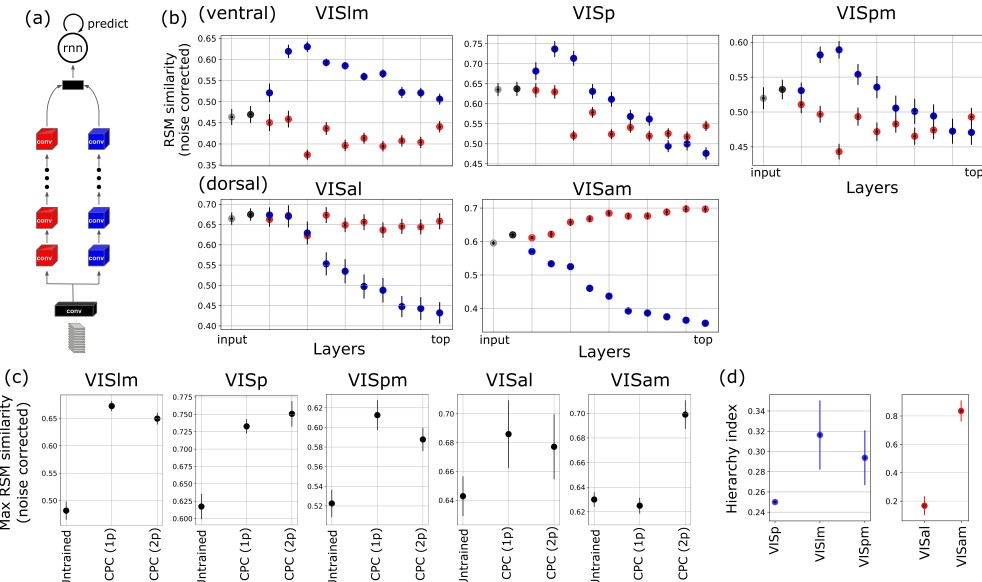

Figure 3: Representational Similarity Analysis between all the visual areas and the ANN trained with CPC. (a) The schematic of the ANN architecture with two pathways (ResNet-2p) used as the backbone of CPC. (b) Representational similarity between all the layers of the ANN with two pathways (trained with CPC) and the ventral (top: VISlm, VISp, VISpm) and the dorsal (bottom: VISal, VISam) areas. (c) The maximum representational similarity values between the ANN and the ventral and dorsal areas. CPC (1p) and CPC (2p) are ResNet-1p and ResNet-2p, respectively, both trained with CPC loss function.(d) Hierarchy index of the ventral (left; in red) and dorsal (right; in blue) areas based on their fit to the ANN. Error bars represent bootstrapped standard deviation.

to the ventral-like pathway from ResNet-2p (Fig. S3). This supports the idea that, in the single pathway model, there is a competition between the two forms of representation that favours ventral-like functions, and which leads to specialized functions in the two-pathway architecture. Using architectures with more than two parallel pathways also does not improve representational similarities (Fig. S5).

The hierarchy index values for the ventral and dorsal areas are shown in Figure 3d. The hierarchy index for every area is calculated based on the model pathway that aligns best with that area in terms of representation similarity (blue pathway for VISlm, VISp, VISpm, and red pathway for VISam, VISal). The hierarchy indices are shown separately for the ventral and dorsal pathways. Similar to the results with ResNet-1p for the ventral pathway, VISp has a lower hierarchy index ($0.25 \pm 0$) than VISlm ($0.31 \pm 0.05$) and VISpm ($0.29 \pm 0.07$). For the dorsal pathway, VISam has a larger hierarchy index than VISal (VISam: $0.82 \pm 0.16$ vs. VISal: $0.19 \pm 0.15$) which is also consistent with the anatomical/functional hierarchy index (Fig. 1c).

In terms of the predictive loss function, the performances of ResNet-1p and ResNet-2p are not significantly different (top-3 accuracy for ResNet-1p: 94.64 (0.68) and ResNet-2p: 93.472 (0.83)). However, it is worth noting that ResNet-2p has, in total, fewer parameters than ResNet-1p (ResNet-1p: 435k vs ResNet-2p: 285k). Therefore, considering the lower capacity of ResNet-2p and its similar predictive performance to ResNet-1p, we can conclude that the inductive bias of parallel architecture could help in predictive processing.

### 4.4 Supervised learning of action recognition with parallel-pathways is not sufficient for dorsal match

The inductive bias of an architecture with parallel pathways combined with the spatiotemporal dynamics of the video data could be enough to trigger the emergence of both ventral and dorsal-like representations, as defined in the previous section, regardless of the loss function used. To understand the role of the loss function, we trained the same ResNet-1p and ResNet-2p backbones with a

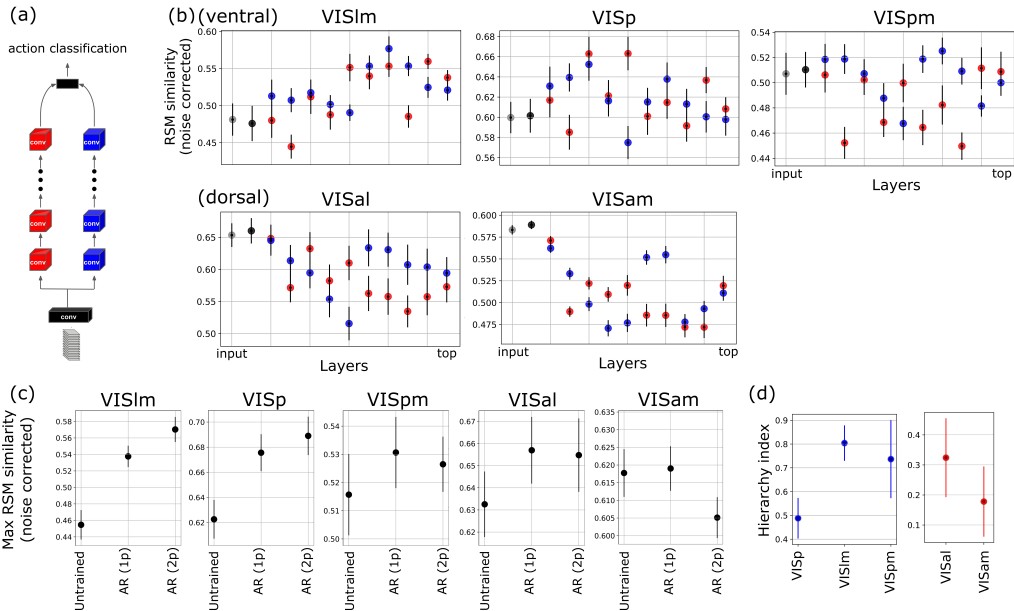

Figure 4: Representational Similarity Analysis between all the visual areas and the ANN trained with supervised action recognition loss function. (a) Representational similarity between all the layers of the ANN with two pathways (trained with action recognition objective) and the ventral-like (top: VISlm, VISp, VISpm) and the dorsal-like (bottom: VISal, VISam) areas. (b) The maximum representational similarity values between the ANN and the ventral and dorsal areas. AR (1p) and AR (2p) are ResNet-1p and ResNet-2p, respectively, both trained with action recognition loss function. (c) Hierarchy index of the ventral (left; in red) and dorsal (right; in blue) areas based on their fit to the ANN. Error bars represent bootstrapped standard deviation.

supervised action classification loss on the UCF101 dataset (Fig. 4a). Based on the representation similarity plots in Figure 4b, both blue and red pathways seem to learn similar representations that are more ventral-like (also see Fig. S2). The maximum RSM similarity values in Figure 4c show that both ResNet-1p and ResNet-2p reach similarity values higher than the untrained ANN for ventral-like areas, but for the dorsal areas, and specifically VISam, neither architecture achieves good representational similarity. The low performance of the action classification models here could not be due to the low spatial resolution of the training videos as our comparisons with ANNs pretrianed with higher spatial resolution videos also show similar results (see supplementary section F).

The hierarchy indices calculated using the ANN trained with an action classification objective (Fig. 4d), reproduce the CPC results (Fig. 3d and 2d) for ventral-like areas, and roughly match the anatomical/functional hierarchy index. However, the action classification model fails to predict the hierarchical organization of the dorsal areas, which is to be expected given the model's poor alignment with these areas. Overall, these findings demonstrate the importance of the CPC loss function for learning both ventral and dorsal-like representations across the ResNet-2p architecture.

### 4.5 Functional specialization of the ventral and dorsal-like pathways in a CPC trained model

The ventral pathway is responsible for object and scene-based tasks [28], while the dorsal pathway is responsible for motion-based tasks [3, 47, 48, 54]. Based on our knowledge of the functional specialization of the two pathways in the real brain, we run a linear evaluation on the two pathways of ResNet-2p trained with CPC on two downstream tasks: (1) object categorization (CIFAR10 dataset) and (2) motion discrimination with random dot kinematograms (RDKs) (see Fig. 5a). In vision neuroscience, RDKs have been commonly used to characterise motion representation in the dorsal pathway [4]. We use this stimulus to evaluate the CPC-trained ResNet-2p (the red pathway in Fig. 3) on motion direction discrimination (four directions: up, down, left, right). Figure 5b shows the results for the two pathways, as well as for an untrained ResNet. As expected based on the comparisons with

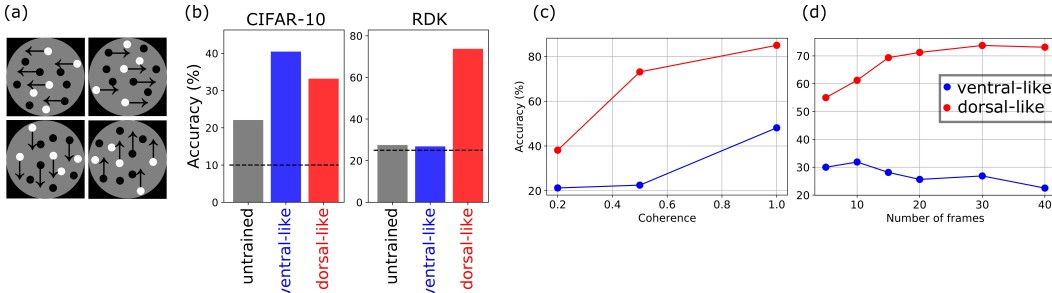

Figure 5: Random dots motion discrimination task. (a) Schematics of the RDK task. (b) The performance of the two pathways and an untrained ResNet on the object categorization (CIFAR-10) and motion discrimination (RDK) tasks. The black dashed lines show the chance levels for the two tasks. (c) The accuracy of the dorsal-like and the ventral-like paths for different levels of the random dots coherence. (d) The accuracy of the two pathways for different number of frames of the random dots stimulus.

mouse visual areas, the ventral-like pathway is better in object categorization, while the dorsal-like pathway outperforms the ventral-like pathway in motion discrimination. Therefore, in addition to their better fits to neural data, the two pathways are indeed more ventral-like and dorsal-like according to their ability to support downstream tasks.

As previously stated, decreasing the dots coherence (increasing spatial noise) makes the random-dots task more difficult. In real brains, dorsal areas can average out noise and extract motion signals by integrating motion over both time and space [46, 60]. It is thought that this is key to the importance of dorsal areas for RDK tasks. We measured ResNet-2p ventral-like and dorsal-like pathway performances for different coherence levels of random-dots (Fig. 5c). Ventral-like performance was dramatically reduced when the coherence was lowered from $100\%$ to $50\%$. In contrast, the decrease in dorsal-like pathway performance was much smaller, demonstrating the dorsal-like pathway's ability to spatially integrate motion, which is consistent with our expectations of dorsal areas. Indeed, the dorsal-like pathway can still achieve just under $40\%$ accuracy even with $20\%$ coherence. We also measured ventral-like and dorsal-like pathway performances for different numbers of frames of the random-dots stimuli at $50\%$ coherence (Fig. 5d). The dorsal-like pathway's performance improved significantly when the number of frames was increased, illustrating the ability of the dorsal-like pathway to integrate motion information over time. On the other hand, increasing the number of frames did not increase ventral-like pathway accuracy, but rather reduced it to around chance level. This shows that unlike the dorsal-like pathway, the ventral-like pathway of ResNet-2p does not integrate motion information over time.

## 5   Discussion

In this paper, we showed that self-supervised learning with CPC can produce representations that are more analogous to mouse visual cortex than either simple models or ANNs trained in a supervised manner. Furthermore, we showed that CPC applied to an architecture that has two parallel pathways can model both the ventral and dorsal areas of mouse visual cortex. The downstream tasks of object recognition and motion discrimination also support the ventral-like and dorsal-like representations of the two pathways in the model. Our experiments with supervised training on action classification indicated that the two-pathway architecture and video dataset are necessary but not sufficient for learning both types of representations, showing an interaction between the self-supervised objective function and the architecture. This finding shows that self-supervised predictive learning is a required component of our model for obtaining both ventral- and dorsal-like representations. Our observation that supervised action classification cannot generate dorsal representations is in contradiction with a previous fMRI study [21]. The different results of the two studies boil down to the different data modalities used in the two studies: two-photon calcium imaging in our study and fMRI in [21]. Capturing the responses of the dorsal areas elicited by natural videos with high temporal dynamics requires neuronal recordings with high temporal sampling rate. Therefore, we hypothesize that some aspects of dorsal representations of movement would not be reflected in the fMRI data, which

could bias the fMRI-based analysis toward more static and lower temporal frequency features of the stimulus and neuronal responses (e.g. very slow-varying features; [61]).

Even though our results demonstrated the importance of self-supervised predictive learning for generating ventral- and dorsal-like representations, it is important to note that our conclusions are limited to the supervised loss functions that we included in our comparisons (i.e. object categorization and action recognition). We acknowledge that a different combination of more ecologically relevant supervised loss functions might be sufficient for learning ventral and dorsal representations in ANNs.

Learning representations of input data (images, videos, etc.) that are invariant to certain data augmentations (e.g. rotation, cropping, etc.) is a common goal of modern self-supervised learning methods. In models such as SimCLR [7] and BYOL [20], the augmentations are engineered for learning the most appropriate representations for downstream tasks. In CPC, however, the augmentations are inherent to the data being used. As noted above, predicting the next frame in a movie requires two different invariances: (1) invariance to motion, but selective for shape, and (2) invariance to shape, but selective for motion. Our results suggest that these two types of invariances are mutually exclusive, which can explain both the need for two separate pathways to get good matches to both ventral and dorsal areas and the inverse relationship between ventral-likeness and dorsal-likeness of the learned representation (see section D and Fig. S4 in supplementary materials). Thus, our results suggest that the functional specialization observed in the mammalian brain may be a natural consequence of a predictive objective applied to an architecture with two distinct pathways.

## 6 Limitations

One limitation of this work is the lack of comparisons with ANNs that are trained with other predictive loss functions, such as PredNet [38]. Other self-supervised video-based learning models (for example, see [14]) that do not optimize a predictive loss function could also be compared with CPC in terms of matching the representations of mouse visual cortex. Another limitation concerns the backbone architectures that we used in this study. Different parameters of the architectures (such as the number of residual blocks, the number of layers before the split and after the merger of the two pathways in in ResNet-2p, etc.) could be searched more thoroughly to determine the optimal setting for modeling mouse visual cortex. Furthermore, training ResNet-2p with CPC on a synthetic video dataset in which the motion and shape contents of the videos can be controlled could demonstrate more directly that the two pathways of ResNet-2p learn motion-invariant shape selectivity and shape-invariant motion selectivity, respectively.

## Broader Impact

ANNs can serve as a framework for understanding brains [50], as demonstrated here. This understanding would be based on finding the loss functions, architecture, and learning rules that best capture brain representations. Technologies that directly or indirectly interface with the brain, such as brain machine interfaces, can benefit from an *in silico* model of the brain. ANNs, if being used as such models, can facilitate designing and optimizing these technologies. However, the downside is that the ANN models of the brain are prone to the same limitations encountered by other ANNs, such as adversarial attack or implicit bias. These limitations can potentially leak into the applications in which these models would be used with human subjects.

## Acknowledgments and Disclosure of Funding

We thank Iris Jianghong Shi, Michael Buice, Stefan Mihalas, Eric Shea-Brown, and Bryan Tripp for helpful discussions. We also thank Pouya Bashivan and Alex Hernandez-Garcia for their suggestions on the manuscript. This work was supported by a NSERC (Discovery Grant: RGPIN-2020-05105; Discovery Accelerator Supplement: RGPAS-2020-00031), Healthy Brains, Healthy Lives (New Investigator Award: 2b-NISU-8; Innovative Ideas Grant: 1c-II-15), and CIFAR (Canada AI Chair; Learning in Machine and Brains Fellowship). CCP was funded by a CIHR grant (MOP-115178). This work was also funded by the Canada First Research Excellence Fund (CFREF Competition 2, 2015-2016) awarded to the Healthy Brains, Healthy Lives initiative at McGill University, through the Helmholtz International BigBrain Analytics and Learning Laboratory (HIBALL).

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
