# Supplementary Material

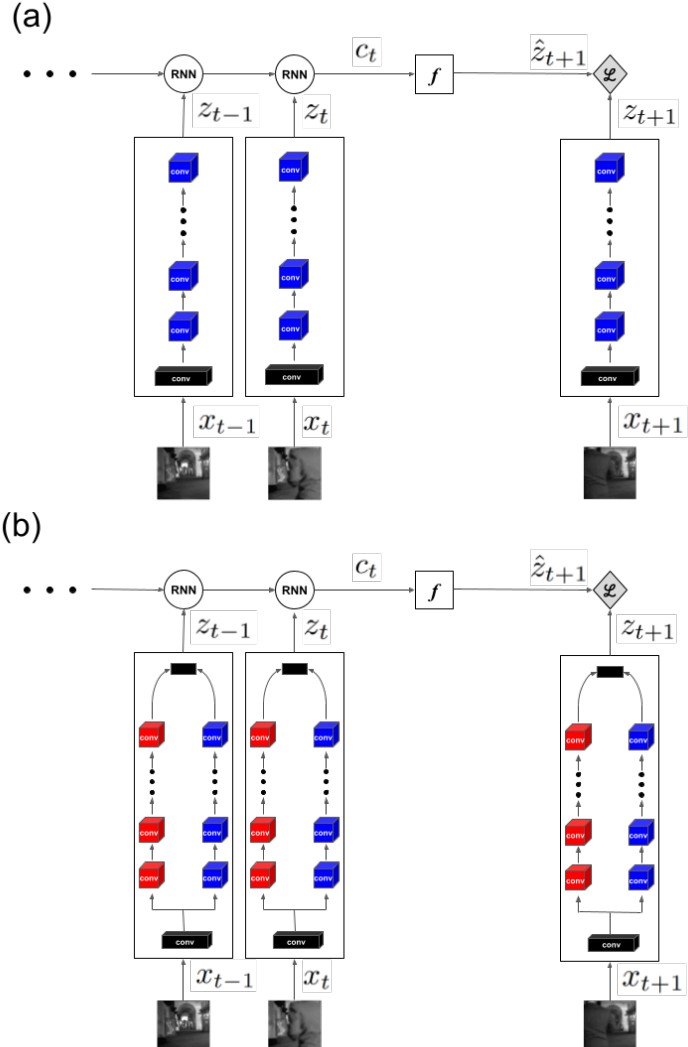

Figure S1: The schematic of Contrastive Predictive Coding model with (a) ResNet-1p, and (b) ResNet-2p backbone architectures. The present and past frames of the video ($x_t, x_{t-1}, ...$) are given as input to the 3D ResNets. The output of the ResNets at each time point ($z_t, z_{t-1}, ...$) are then passed to a recurrent neural network (RNN) which generates a context variable at time $t$ ($c_t$). The context variable, $c_t$, is then used to predict the latent state of the next frame ($\hat{z}_{t+1}$) via a single layer MLP ($f$). The predicted next frame along with the latent representation of the correct next frame (positive sample, $z_{t+1}$) and incorrect samples (negative examples, not shown here) are then fed to the loss function (see equation 1).

# A  Supplementary methods

## A.1  RSA noise ceiling

To estimate the noise ceiling of $RSM$ similarity (the maximum $RSM$ similarity we can reach for a given area), we randomly split the neuronal responses into two groups, and calculate two different $RSM$s based on each split. We then calculate the similarity between the $RSM$ of the splits (as explained in section 3.2), and the similarity value gives us an estimate of the best possible match (i.e. the noise ceiling) for the recordings of that brain area. This process is repeated 100 times to also obtain a measure of variability for the estimated noise ceiling. The diagonal values of the matrix in Figure 1b shows the average noise ceiling values for every area.

## A.2  Contrastive predictive coding

As noted in section 3.3, the CPC loss function relies on predicting the future latent representations of a video sequence, given its present and past representations (Fig. S1). Specifically, a block of $N$ frames ($x_t \in \mathbb{R}^{N \times W \times H \times C}$; $W, H$ are the spatial dimensions, $C$ is the number of input channels) are passed through a backbone architecture, here a 3D convolutional neural network (CNN), and the CNN output ($z_t \in \mathbb{R}^{1 \times W' \times H' \times D}$; $W', H'$ are the spatial dimensions, $D$ is the number of output channels) is given to a recurrent neural network (RNN). The RNN aggregates the latent variables of $S$ blocks of frames ($z_{t-S+1}, ..., z_t$; here $S = 5$), and generates the context variable $c_t$ which is then used to predict the future $T$ steps of the latent variables ($\hat{z}_{t+1}, ..., \hat{z}_{t+T}$; here $T = 3$). Importantly, the prediction is done in the latent space (*i.e.* the CNN output), and not in the pixel space. Specifically, CPC optimizes the following contrastive loss function:

$$\mathcal{L} = -\sum_i \log \left[ \frac{\exp(\hat{z}_i^T . z_i)}{\sum_{j \neq i} \exp(\hat{z}_i^T . z_j)} \right] \tag{1}$$

where, $i$ and $j$ denote the $i^{th}$ and $j^{th}$ time points. Minimizing the above contrastive loss function maximizes the similarity of the predicted latent variable ($\hat{z}_i^T$) and the true future latent state ($z_i$; AKA positive pairs), and minimizes its similarity with incorrect latent states ($z_j$ for $j \neq i$; AKA negative pairs).

## A.3  Other models

Two models are used as baselines: 3D Gabors, that were shown to be an acceptable model of the primary visual cortex in cats and primates [55, 29], and an untrained, randomly initialized 3D ResNet. The Gabor model comprised of 3D Gabor filters, spanning 8 motion directions (equally spaced between 0 and 315 degrees), 3 phases (-1, 0, 1), and 5 spatial scales. The randomly initialized model has its weights selected by matching the mean and standard deviation of synaptic weights in the trained models. These two baseline models are expected to be surpassed, in terms of alignment with brain representations, by any trained ANN that is using a relevant loss function for the visual system (meaning that the loss function captures something relevant about how either evolution or learning shape the brain). In addition to the baseline models, we also compare the CPC trained ANNs with ANNs that are trained with two other supervised loss functions: supervised object categorization (ImageNet dataset) and supervised action recognition (UCF101 dataset). We consider the ANNs trained with object categorization simply because they have been the standard in studies of the ventral pathway. We also use supervised action recognition (with UCF101 video dataset) in order to ensure that we have a fair comparison with CPC, which is being trained with dynamic videos. Importantly, both CPC and supervised action recognition use the UCF101 dataset, which makes the loss function the only difference between these two models. In order to show that the spatiotemporal dynamics of the video data is important, we also compared CPC with SimCLR, a self-supervised learning model trained on static images [7]. All the models are compared based on their representation alignment with different areas of mouse visual cortex (see section 3.2), and by using downstream tasks (see section 3.3).

## A.4  Datasets

For training the deep ANNs, we use the UCF101 dataset. UCF101 is a dataset of 13320 short video segments from 101 action categories collected from YouTube [56]. We use this dataset for both

self-supervised and supervised training of the ANNs. In the case of supervised training, the ANNs are trained to categorize actions in the videos. However, the UCF101 videos are not used for comparing representations between brain areas and the ANNs. For this, we use the same videos that the Allen Brain Observatory presented to the mice. Both datasets (UCF101 and Allen Brain Observatory videos) were normalized (with mean and standard deviation calculated across each dataset) and downsampled to $64 \times 64$ to account for the low spatial resolution of mouse retina.

## A.5 Linear evaluation tasks

As noted in section 3.3, we examine the two pathways of our trained ResNet-2p on two downstream tasks: object categorization and motion discrimination, which are supported in the real brain by the ventral and dorsal pathways, respectively. For each downstream task, the weights of the trained pathways are frozen and a linear classifier is trained on the final convolutional layer of each pathway. We use the CIFAR-10 dataset for the object categorization task [36]. For the motion discrimination task, we use random-dot kinetograms (dot density $= 2.5\%$, dot size $= 2\ pixels$, speed $= 5\ pixels/frame$). In every sample of the stimulus, a portion of dots move coherently in one of four directions (up, down, right, left), and the rest move completely randomly. The linear classifier is trained to detect the principal direction of motion. The models are evaluated on 3 motion coherence levels ($20\%$, $50\%$, $100\%$) and 6 stimulus durations ($5, 10, 15, 20, 30, 40$ frames).

## A.6 Code availability

The codes repository and the pretrained models are available at: `https://ventral-dorsal-model.netlify.app/`.

Table S1: An instantiation of the ResNet-1p and ResNet-2p architectures. The dimensions of kernels are shown by $T \times S^2, C$ for temporal, spatial, and channel dimensions. The parameters of the two pathways of ResNet-2p are shown separately in red and blue colors. Output sizes are also denoted by $T \times S^2$ for temporal and spatial dimensions. The concatenate layer is only included in ResNet-2p where the two pathway outputs are concatenated along the channel dimensions. The sequence of res$_3$ and res$_4$ residual blocks repeats 4 times. In total, there are 10 residual blocks in both ResNet-1p and ResNet-2p.

| Stage | ResNet-1p | ResNet-2p | Output size |
|---|---|---|---|
| data layer | | | $5 \times 64^2$ |
| conv$_1$ | $5 \times 7^2, 256$ stride $1 \times 1^2$ | $5 \times 7^2, 256$ stride $1 \times 2^2$ | $5 \times 32^2$ |
| pool$_1$ | $1 \times 3^2$ stride $1 \times 2^2$ | $1 \times 3^2$ stride $1 \times 2^2$ | $5 \times 16^2$ |
| res$_1$ | $\begin{bmatrix} 1 \times 1^2, 256 \\ 1 \times 3^2, 32 \\ 1 \times 1^2, 128 \end{bmatrix}$ | $\begin{bmatrix} 1 \times 1^2, 128 \\ 1 \times 3^2, 16 \\ 1 \times 1^2, 64 \end{bmatrix}, \begin{bmatrix} 1 \times 1^2, 128 \\ 1 \times 3^2, 16 \\ 1 \times 1^2, 64 \end{bmatrix}$ | $5 \times 16^2$ |
| res$_2$ | $\begin{bmatrix} 3 \times 1^2, 128 \\ 1 \times 3^2, 32 \\ 1 \times 1^2, 128 \end{bmatrix}$ | $\begin{bmatrix} 3 \times 1^2, 64 \\ 1 \times 3^2, 16 \\ 1 \times 1^2, 64 \end{bmatrix}, \begin{bmatrix} 3 \times 1^2, 64 \\ 1 \times 3^2, 16 \\ 1 \times 1^2, 64 \end{bmatrix}$ | $5 \times 16^2$ |
| res$_3$ | $\begin{bmatrix} 1 \times 1^2, 128 \\ 1 \times 3^2, 32 \\ 1 \times 1^2, 128 \end{bmatrix}$ | $\begin{bmatrix} 1 \times 1^2, 64 \\ 1 \times 3^2, 16 \\ 1 \times 1^2, 64 \end{bmatrix}, \begin{bmatrix} 1 \times 1^2, 64 \\ 1 \times 3^2, 16 \\ 1 \times 1^2, 64 \end{bmatrix}$ | $5 \times 16^2$ |
| res$_4$ | $\begin{bmatrix} 3 \times 1^2, 128 \\ 1 \times 3^2, 32 \\ 1 \times 1^2, 128 \end{bmatrix}$ | $\begin{bmatrix} 3 \times 1^2, 64 \\ 1 \times 3^2, 16 \\ 1 \times 1^2, 64 \end{bmatrix}, \begin{bmatrix} 3 \times 1^2, 64 \\ 1 \times 3^2, 16 \\ 1 \times 1^2, 64 \end{bmatrix}$ | $5 \times 16^2$ |
| concatenate | — | | $5 \times 16^2$ |

## B    Representational similarity between the two pathways of ResNet-2p

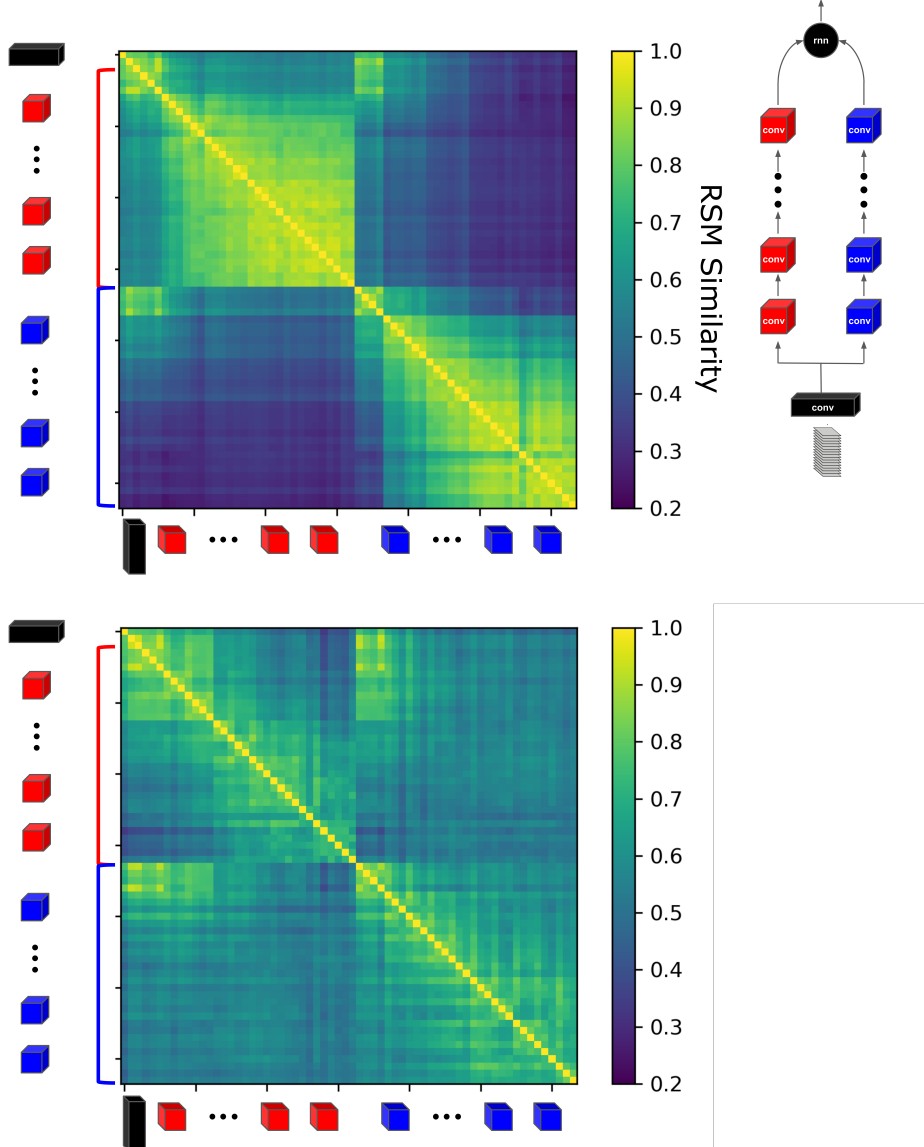

Figure S2: Representational similarity between and within the two pathways of ResNet-2p trained with CPC (top) and action recognition (bottom) loss functions.

## C   Representational similarity between the two pathways of ResNet-2p and one pathway of ResNet-1p

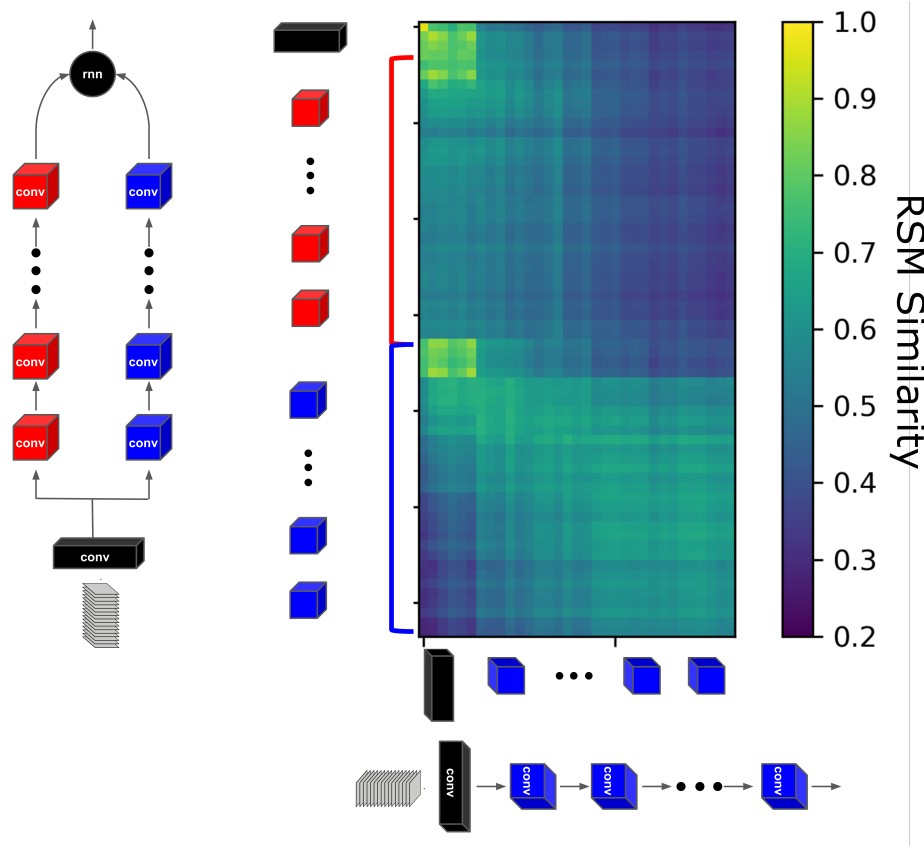

Figure S3: Representational similarity between ResNet-1p (columns in the figure) and ResNet-2p (rows in the figure) trained with CPC loss function. The blue and the red pathways in ResNet-2p schematic represent the ventral- and the dorsal-like pathways, respectively. As can be seen in the figure, the lower part of the rectangle has higher values than the upper part showing that ResNet-1p has more similar representations to the ventral-like pathway (blue in ResNet-2p schematic) than the dorsal-like pathway (red in ResNet-2p schematic).

## D  Representational similarity during training

To gain an insight into the development of the dorsal- and ventral-like pathways, we quantify the similarity of each pathway in ResNet-1p and ResNet-2p to dorsal and ventral areas during training with CPC. For that purpose, we use the maximum representation similarity with VISam and VISlm (the most dorsal and ventral areas, respectively) as the dorsal-score and ventral-score, respectively. The scores are normalized between 0 and 1 and shown in Figure S2, for both ResNet-1p (left) and ResNet-2p (right). The results suggest that there is a competition between dorsal- and ventral-like representations in the one-pathway architecture. An increase in ventral score leads to a decrease in the dorsal score in ResNet-1p, which is reflected in the correlation value of $r = -0.54$ between the two scores. However, the second pathway in the ResNet-2p model decreases the anti-correlation between the two scores to $r = -0.18$. Indeed, by assigning the two representations to separate pathways in ResNet-2p, the two representations become partially independent. This result suggests that a two-pathway architecture can learn both ventral- and dorsal-like representations because it can partially decouple these two competing forms of representation.

For 10 random initialization seeds, we do get the dorsal/ventral split in 7 seeds (average D-score = 0.714 , average V-score = 0.663). In 1 seed, both pathways are more dorsal-like (D-score = 0.688, V-score = 0.504) and in 2 seeds, more ventral-like (average D-score = 0.620, average V-score = 0.664). Also, the seeds with better dorsal/ventral splits across the two pathways reach higher average predictive performance at the end of training (top-3 accuracy with dorsal/ventral split: 93.88 (1.02), without dorsal/ventral: 91.80 (1.17)).

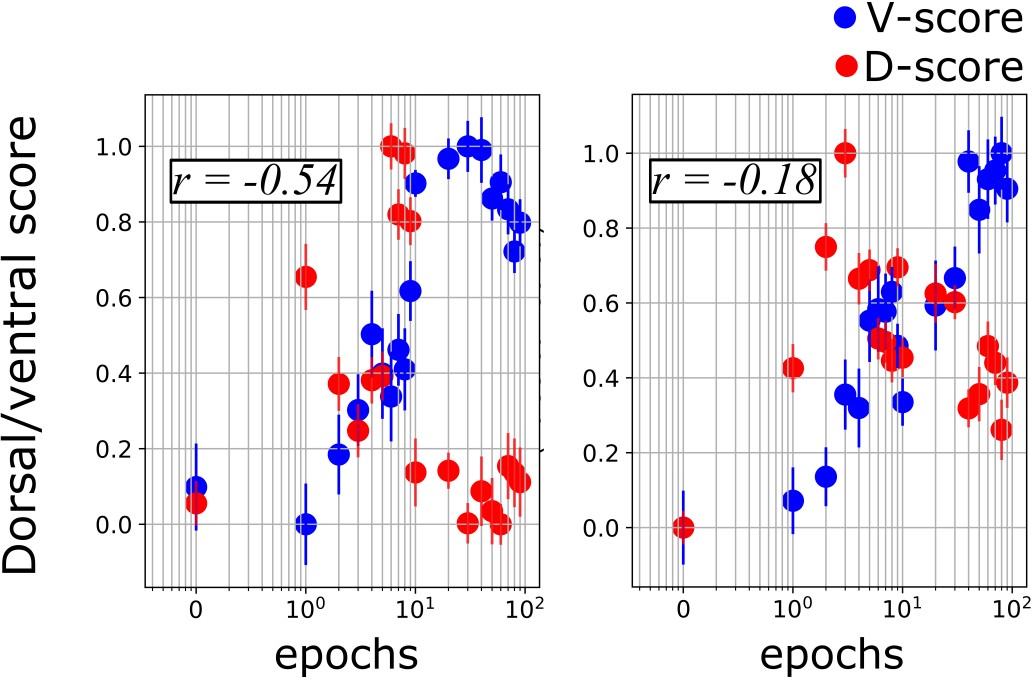

Figure S4: Ventral (red) and dorsal (blue) scores during training for the architectures with one (left) and two (right) pathways. $r$ shows the correlation coefficient between the changes in dorsal and ventral scores during training.

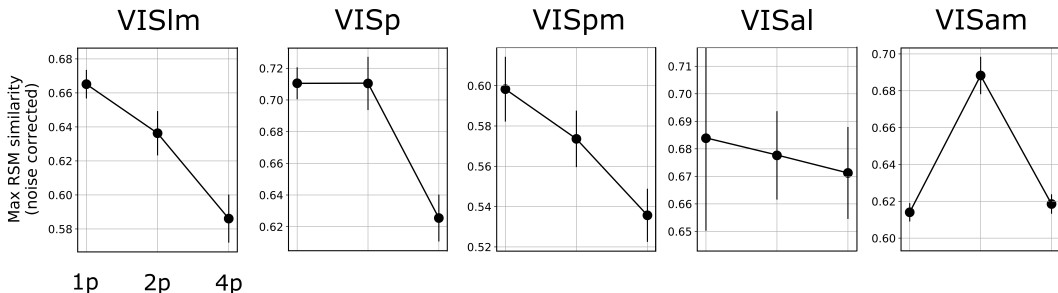

Figure S5: Maximum representational similarity for ResNet-1p, ResNet-2p, and ResNet-4p trained with CPC.For all the areas of mouse visual cortex, increasing the number of pathways from 2 to 4 decreased the representational similarity.

# E  Dorsal-like specialization in the one-pathway model

Although the one-pathway architecture trained with CPC showed more similar representations with the ventral pathway in mouse visual cortex (Fig. 2), we can still ask explicitly if there is any learned dorsal-like specialization in ResNet-1p. To address this question, we take two approaches: (1) we measure the performance of ResNet-1p on the two downstream tasks, CIFAR-10 and RDK, as explained in section A.5, and (2) predicting the responses of the artificial neurons in ResNet-1p based on the responses of the neurons of the most ventral (VISlm) and the most dorsal (VISam) areas of mouse visual cortex. For (2), particularly, we take every artificial neuron in ResNet-1p, and fit a linear regression model to the neurons' responses. We use 10-fold cross-validation to evaluate the linear fits. Neurons are labeled as dorsal-like if the dorsal neurons from VISam are better predictors of that artificial neuron's responses than VISlm neurons.

The results of the two approaches are shown in Table S2 below. The linear evaluation results show that ResNet-1p is better than both ventral-like and dorsal-like pathways of ResNet-2p in object categorization, supporting the observation that ResNet-1p representations are more ventral-like. For the motion discrimination task, ResNet-1p is better than the ventral-like pathway of ResNet-2p, but worse than the dorsal-like pathway of ResNet-2p. This observation implies that ResNet-1p has some dorsal-like specialisation, but separating the two pathways in the architecture enhances dorsal-like specialisation in one of the pathways.

Consistent with the results of the downstream tasks, the linear regression approach also show that around 39% of ResNet-1p neurons are dorsal-like. When compared to the percentage of dorsal-like neurons in each pathway of ResNet-2p, we can see that using an architecture with parallel pathways enhances and isolates ventral and dorsal specializations in each pathway: the dorsal-like pathway has a significantly higher number of dorsal-like neurons compared to the ventral-like pathway and ResNet-1p.

Table S2: The percentage of dorsal-like neurons and the performance on downstream tasks (CIFAR-10 object categorization and RDK motion discrimination) for ResNet-1p and each pathway of ResNet-2p

| Model | % dorsal-like neurons | CIFAR-10 | RDK |
|---|---|---|---|
| ResNet-1p | 38.96 (1.30) | 58.75 | 43.90 |
| ResNet-2p (v-like) | 17.00 (1.05) | 38.12 | 26.875 |
| ResNet-2p (d-like) | 60.09 (3.73) | 33.20 | 73.75 |

# F Comparisons with pretrained high resolution action recognition models

The ANNs trained on supervised action classification, similar to the CPC-trained models, were trained with low spatial resolution videos ($64 \times 64$; UCF101 dataset) to account for the low spatial resolution of mouse retina. The low resolution of the video data might affect the quality of the learned representations, especially in supervised learning. Therefore, in this section, we compare CPC with an ANN pretrained with higher resolution action classification datasets ($112 \times 112$; Kinetics-400 dataset). We also compare the models with the SlowFast network: a two-stream architecture pretrained on action classification. However, unlike the two-pathway architecture used in our study, the two pathways of the SlowFast network are not identical. By choosing different stride sizes, kernel sizes, and input sampling rates, one pathway (the Slow pathway) was designed for learning representations of more static information, while the other pathway (the Fast pathway) was designed for capturing rapidly changing motion (see [13] for details).

The results of these comparisons are shown in Figure S6. First, we can see that the pretrained action classification model (AR (high res)) shows a lower representational similarity to all the brain areas compared to CPC, consistent with the results presented in Figure 4. Therefore, the lower performance of the supervised models, as reported in section 4.4, cannot be explained by the low resolution of the input videos that we used in the training.

Second, the SlowFast network performs at the same level as the action classification model for the two most ventral areas (VISlm and VISp) and the most dorsal area (VISam). Therefore, training an ANN with two parallel pathways on supervised action classification does not lead to learning ventral- and dorsal-like representations, consistent with the results in section 4.4 and Figure 4. However, the SlowFast network shows the highest representational similarity with two of the visual areas: VISpm (ventral) and VISal (dorsal). We can attribute the superior performance of the SlowFast network, in the case of these two areas, to the inductive biases used in this model. In particular, the temporal sampling rate of the input videos for training the Fast pathway of the SlowFast network was higher than the input sampling rate we used for training our models. While, in this study, we focused on the role of loss functions and number of pathways in the architecture, these results highlight the importance of other architectural hyperparameters in modeling different visual areas. Importantly, these results suggest that different visual areas may be best modeled with different architectural and input hyperparameters.

The pretrained action recognition model and the SlowFast network were both trained on Kinetics-400 which is a much larger dataset compared to UCF101 that we used in this study (650k samples in Kinetics-400 vs 13k samples in UCF101). Therefore, we should bear in mind that the training dataset is a confounding factor here when comparing the pretrainied models with CPC.

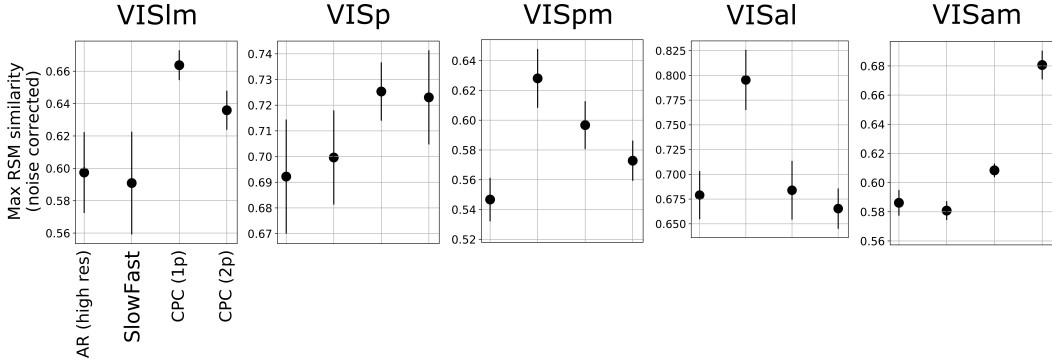

Figure S6: Maximum representational similarity for models trained on supervised action classification with high resolution videos (AR (high res)), the SlowFast network, and ResNet-1p and ResNet-2p trained with CPC.