# OpenReview forum: "The functional specialization of visual cortex emerges from training parallel pathways with self-supervised predictive learning"
_NeurIPS.cc/2021/Conference — NeurIPS 2021 Spotlight_

### Official Review · Reviewer_oMSB · 2021-07-15

**Rating:** 7
**Confidence:** 4

**Summary:**

The authors show that a two-stream 3d ResNet architecture trained using a contrastive predictive coding (CPC) objective develops a representation that resembles the ventral/dorsal stream split in visual cortex. Although I do not buy into all claims of the paper, I am generally very supportive of it. The authors make their main point and the paper is well written. With some adjustments of the claims it will be a great paper.


### Update after discussion period

Given the results from the control experiments reported by the authors, I support publication of the paper and increased my score to 7. I do ask the authors to include those results in the final version of the paper.

**Limitations And Societal Impact:**

Yes

**Main Review:**

## Strengths
+ Simple but innovative idea
+ Demonstrates a single network architecture and objective function can generate a ventral/dorsal split as in visual cortex
+ Well written and easy to read

## Weaknesses
- Unclear if claimed superiority of CPC over supervised pre-training holds
- Unclear how robust the emergence of two streams is


## Detailed comments on weaknesses

### 1. Comparison to supervised pre-training

Using the CPC objective for training, the authors demonstrate nicely that the two-stream architecture results in more dorsal-like representations in its second stream than the one-stream architecture. However, I am concerned about their claim that the CPC objective is necessary.

- The supervised baseline might be problematic due to the low resolution of the videos. What can be recognized at 64x64 pixels? What's the performance of the classifier on the task it was trained on? If the task cannot really be solved, I'd be worried that the network simply doesn't learn useful representations in the first place, because the semantic labels are not much related to the image information present at the low resolution. Following the authors' reasoning, supervised pre-training should perform well on the ventral stream areas. However, predictive performance of the supervised model is quite weak for all brain areas. This result seems to suggest that comparison to the supervised model is not very meaningful. Wouldn't it make more sense to compare to a well-trained action classification net that uses higher-resolution movies and take care of upsampling the videos shown to the mouse accordingly?

- (Related to the previous point.) Previous work [20] has demonstrated dorsal-like hierarchical representations in a 3d CNN trained on UCF101 (Tran et al. 2014). The authors cite that paper in their related work section, but do not further comment on it. Given that their claim that supervised training is not sufficient goes directly against published work, I would have expected at least some kind of discussion/explanation, and I think the burden of proof for such a claim is on the authors.

- There is an existing two-stream video recognition architecture (SlowFast networks: https://arxiv.org/abs/1812.03982), which the authors don't seem to be aware of (at least it's not cited in the related work). Given that it's publicly available, trained in a supervised fashion and features two streams, I think it should be included in the comparisons in Figs. 2–4.

In summary, if the authors want to make the point that the CPC objective is necessary and supervised training does not work, they need to provide substantially more evidence for this claim. An alternative solution could be to remove this claim and focus on making the point that CPC + two-stream architecture are a sufficient condition for achieving a ventral/dorsal split, but not claim anything about necessity.


### Robustness of the emergence of two streams

As I understand the paper, the authors trained exactly one instance of the two-stream architecture using CPC and are basing their claims on this single instance. I wonder how robust this result is, though. Do these two streams always emerge or were the authors "lucky"? It would be good to have at least a second, independent verification with a different seed – and, perhaps, slightly different hyperparameters – to verify that the finding is at least somewhat robust.


### Minor comments

- How was the input resolution for the ANNs determined? Depending on the resolution one picks, different layers may be optimal simply because of the receptive field sizes of neurons. It would be good to clarify this point.

- Fig. 2+3: What do the error bars depict?

- Fig. 2–4: Why are the gray points not identical? I thought they depict the RSM similarity of the  input (pixels) with the respective brain area? Those shouldn't depend on the model, should they? The same goes for the untrained model in Fig. 3+4(c): Shouldn't the two also be identical? After all, it's the same architecture.

**Time Spent Reviewing:**

3

---

> ### Author Response · Authors · 2021-08-10
> **Response to Reviewer oMSB**
>
> We thank the reviewer for being supportive of our work and for the thoughtful and constructive comments. We will do our best to address their comments, and revise the paper accordingly. Please see our detailed response to the comments below, including our proposed changes to the paper (we are not allowed to upload a revised paper at this time):
>
>
> **1- Low spatial resolution of the training videos**
>
> Our supervised-trained model on the action recognition task reached around 75% training and 50% validation top-1 accuracy which is well above the chance level of this task (~ 1%). Therefore, we can confidently say that the model was able to solve the task partially which is what we would expect from a model with low input resolution. We will include a table with these results in the supplemental material of the revised manuscript. That is, the model learned to discriminate classes of samples that could be differentiated at low resolution but not the classes that required higher input resolution. And, this is also what we would expect from the visual capabilities of mice.
>
> In addition, the UCF101-trained model showed reasonable performance on the ventral areas, specifically VISlm and VISp, which suggests that the model was not undertrained (see Figure 4c). It is true, though, that the learned representations of this model were less similar to the mouse data, in comparison to CPC. But, we believe that this is exactly because the classification loss on semantic labels is not a relevant ecological task for mice. The same goes for the model trained on object categorization (Figure 2c). It's worth noting that both action recognition and object classification resulted in equal degrees of representational similarity in the visual cortex of mice. Indeed, we note that a recent concurrent study (Nayebi et al. 2021, biorxiv) also corroborated this finding. In this paper, they showed that self-supervised models are more successful in modeling the representations of the mouse visual cortex than supervised models. In other words, unlike findings in primate studies, supervised training on human-defined semantic labels does not appear to be an ecologically relevant loss function for mice, and the models trained with these labels perform worse than those trained with self-supervised and predictive loss functions in modeling mouse visual representations. That being said, none of these two studies rule out the possibility of a supervised task (other than object categorization and action recognition) that would be more ecologically relevant to mice (e.g. place categorization) and might potentially develop representations more similar to mouse visual cortex. We will clarify in the revised manuscript that our conclusions about supervised training are limited to the supervised tasks we examined, and training conditions included in this paper.
>
> We will also include a comparison with other 3D ResNet models that are pre-trained on high resolution videos. The experimentation with one- vs two-pathway architectures will not be feasible with the pretrained models as they mostly come with a simple, one-pathway architecture. However, as we explain below (see item 3), we will include a comparison with the SlowFast network as a model that is pre-trained on action recognition and has an architecture with two parallel pathways.
>
>
> **2- About Ref [20]**
>
> We agree with the reviewer that, given the common goals of our paper and [20], we should provide some explanation on the apparent contradictions between the results of the two papers. Due to space constraints, we were unable to expand on the results presented in [20].
> In short, we believe that the different results of the two papers boil down to the different data modalities used in the two studies: two-photon calcium imaging in our study and fMRI in [20]. Capturing the responses of the dorsal areas elicited by natural videos with high temporal dynamics requires neuronal recordings with high temporal sampling rate. The moving features and high temporal frequency dynamics of visual stimuli that mainly drive the high frequency responses of the neurons along the dorsal pathway (e.g. neurons with ~10 Hz preferred temporal frequency) could be lost in fMRI data with its slow time-course and complicated BOLD response. As such, we hypothesize that some aspects of dorsal representations of movement would not be reflected in the fMRI data. This would bias the analysis toward more static and lower temporal frequency features of the stimulus and neuronal responses (e.g. very slow-varying features; Wiskott and Sejnowski, 2002). The results of [20] showed that these BOLD responses, that were most likely driven by low temporal frequency and static features of videos due to the nature of fMRI, could be partially explained by an ANN trained on action recognition. We note that this is actually consistent with our observations that ANNs trained on action recognition are mostly driven by the spatial (and not temporal) statistics of input videos (see also Huang et al. 2018, CVPR). We will add some discussion of these matters to the revised manuscript.
>
> **3- SlowFast network**
>
> The SlowFast network is indeed very relevant to our work, and it was also cited in our paper (see Ref 12) as the main inspiration for the backbone architecture of our model.
>
> We will include a comparison with this model in our revised manuscript. However, given the different inductive biases built into the two Slow and Fast pathways of the SlowFast network, the comparison will introduce an additional point into the paper: namely, if we use supervised action recognition loss but add inductive biases to the two pathways, do ventral- and dorsal-like representations emerge in the two pathways? This additional point calls for a more detailed explanation of the results and the motivations behind running this experiment. We will devote a section in the supplementary materials to report the results of the comparisons with the SlowFast network (as well as the pretrained 3D ResNets mentioned above) and elaborate on how it can help us understand the emergence of dorsal/ventral representations in the brain. We will also revise the main text to refer to this additional comparison and the motivation behind it.
>
> **4- Robustness**
>
> We see the reviewer’s point, and we will include the results of a few more random seeds in the revised manuscript.
>
> With regards to hyperparameters, they were optimized for the training tasks, not for increasing representational similarity with the mouse brain (we will be sure to make this very clear in the revised text). Thus, by changing the hyperparameters, we would push the models into a suboptimal training and/or inference regime that could obfuscate the comparisons.
>
> **5- Minor comments**
>
> **How was the input resolution for the ANNs determined? Depending on the resolution one picks, different layers may be optimal simply because of the receptive field sizes of neurons. It would be good to clarify this point.**
>
> This is an important point and we will clarify it in the revised manuscript.
>
> The 64x64 spatial dimensions of the input videos is chosen to roughly mimic the low acuity of mouse vision (see Prusky et al., Vision Research, 2000). We agree that the match between different layers of the ANNs and different brain areas could be affected by the input resolution if the receptive field size, and not the feature selectivities, dictate the matches. But as shown in a previous study (Ref [48] in the paper), the optimal layers for every area is not affected much by the input resolution when using the representational similarity metric that we used in this study. This would not be necessarily true with other measures such as CCA or linear regression.
>
> **Fig. 2+3: What do the error bars depict?**
>
> Every similarity measurement is bootstrapped by repeatedly sampling the population of neurons in our calcium imaging dataset. The error bars show the standard deviation of the bootstrapped measurements. We will clarify this in the methods section.
>
> **Fig. 2–4: Why are the gray points not identical? I thought they depict the RSM similarity of the input (pixels) with the respective brain area? Those shouldn't depend on the model, should they? The same goes for the untrained model in Fig. 3+4(c): Shouldn't the two also be identical? After all, it's the same architecture.**
>
> The very small variability between the gray points (pixel-level representational similarities) in different plots are reflecting the variance in the noise ceiling estimation. Since for each comparison (Figures 2-4) we ran the whole analysis and estimated the neuronal noise ceiling separately, there is a small variability in the noise ceiling estimation that is reflected in the apparent variability of pixel-level representational similarities across figures.
>
> This is also true for the untrained models, but in addition to that, the untrained models were also initiated randomly for every comparison, which introduced a second source of variability into the measurements for the untrained models.

---

> > ### Comment · Reviewer_oMSB · 2021-09-01
> > **Not entirely convinced**
> >
> > Thank you for the response and clarifications. Unfortunately I'm not really convinced.
> >
> > I still think that if you want to make the point that the CPC objective is *necessary* and supervised training does not work, you need to provide substantially more evidence for this claim. An alternative solution could be to remove this claim and focus on making the point that CPC + two-stream architecture are a *sufficient* condition for achieving a ventral/dorsal split, but not claim anything about necessity.
> >
> > Also, there was plenty of time to perform at least some of the control experiments. Without knowing the results it's difficult to know whether they support or contradict the claims of the paper.

---

> > > ### Author Response · Authors · 2021-09-01
> > > **Response to Reviewer oMSB**
> > >
> > > We thank the reviewer for their response. There are two important points that need to be emphasized here: 1) we *do not* claim that supervised learning could never produce dorsal-like representations. We will ensure that this point is clearly explained in the paper. 2) We will highlight in the paper that predictive self-supervised learning is *sufficient* but *not necessary* for dorsal/ventral dual representations in the visual system.
> > >
> > > We have the results of the comparisons with a video model pretrained on higher resolution videos (frame sizes 112 by 112). The model is trained on Kinetics-400 which is a much larger dataset compared to UCF101 that we used in this study (~650k samples in Kinetics-400 vs ~13k samples in UCF101). Therefore, we should bear in mind that the training dataset is a confounding factor here when comparing the pretrainied model with CPC.
> > >
> > > The results of the pretrained video models are close to the results previously reported in the paper. Training on action recognition with higher resolution videos still cannot explain the dorsal areas, similar to the results reported in the paper. Moreover, CPC, in general, shows higher representational similarity with all the areas of mouse visual cortex, which supports the hypothesis that the predictive loss function aids in modeling mouse visual cortex more accurately. Here are the maximum representational similarities (similar to Figure 2c) for CPC and a 3D ResNet pretrained on Kinetics-400:
> > >
> > > VISlm	(**Act Recog**): 0.590 (0.025) &nbsp; &nbsp; (**CPC-1p**): 0.659 (0.008) &nbsp; &nbsp;(**CPC-2p**): 0.628 (0.0125)
> > >
> > > VISp    (**Act Recog**): 0.682 (0.024) &nbsp; &nbsp; (**CPC-1p**): 0.717 (0.011) &nbsp; &nbsp; (**CPC-2p**): 0.722 (0.016)
> > >
> > > VISpm (**Act Recog**): 0.557 (0.014) &nbsp; &nbsp; (**CPC-1p**): 0.609 (0.014) &nbsp; &nbsp; (**CPC-2p**): 0.582 (0.011)
> > >
> > > VISal   	(**Act Recog**): 0.675 (0.023) &nbsp; &nbsp; (**CPC-1p**): 0.679 (0.031) &nbsp; &nbsp; (**CPC-2p**): 0.691 (0.016)
> > >
> > > VISam (**Act Recog**): 0.593 (0.010) &nbsp; &nbsp;  (**CPC-1p**): 0.620 (0.004) &nbsp; &nbsp;(**CPC-2p**): 0.694 (0.010)
> > >
> > >
> > > We have not yet been able to run the comparisons with the SlowFast model, unfortunately. We have encountered several problems while running the code, and it requires more work to generate the results of this comparison. We will definitely include the results in the revised paper. We do not believe that the results of this comparison will affect our conclusions in any way given that, as mentioned before, we believe predictive self-supervised learning is sufficient but not necessary for dorsal/ventral dual representations in the visual system, and we will emphasize this in the revised paper. It is possible that the combination of action recognition loss function and the inductive biases of the Slow and Fast pathways of the SlowFast model generate dorsal-like and ventral-like representations, and this would not contradict with any of our claims in the paper, once we have made these revisions.

---

> > > > ### Comment · Reviewer_oMSB · 2021-09-02
> > > > **That's reassuring**
> > > >
> > > > Thanks, that's indeed reassuring and indeed supports the story. What about the additional training runs? Do you always get two streams?

---

> > > > > ### Author Response · Authors · 2021-09-02
> > > > > **Initialization seeds**
> > > > >
> > > > > Thanks for your response!
> > > > >
> > > > > Regarding additional training runs: in 10 random seeds that we tested, we did get the dorsal/ventral split in 7 seeds (average D-score = 0.714 , average V-score = 0.663). In 1 seed, both pathways were more dorsal-like ( D-score = 0.688 , V-score = 0.504) and in 2 seeds, more ventral-like (average D-score = 0.620 , average V-score = 0.664). Also, the seeds with better dorsal/ventral splits across the two pathways reached lower average predictive loss at the end of training (with dorsal/ventral split: 1.450 (0.036), without dorsal/ventral: 1.573 (0.038) ).
> > > > >
> > > > > Therefore, the dorsal/ventral split is partially dependent on the initialization seed, but it is more likely for it to emerge than not. Moreover, those seeds that allow a dorsal/ventral split to emerge seem to also be better for the predictive training, supporting the idea that the dorsal ventral split helps with the predictive task. Speculatively, this is interesting with respect to the real brain, as real brains are not initialized randomly, and may come wired to promote the emergence of certain types of solutions

---

> > > > > > ### Comment · Reviewer_oMSB · 2021-09-02
> > > > > > **Please include in revision!**
> > > > > >
> > > > > > Thanks for the response! I think this is an important piece of information that should definitely be included in the final version of the manuscript!
> > > > > >
> > > > > > Given these and the other last-minute results, I'm now much more happy to support the paper.

---

> > > > > > > ### Author Response · Authors · 2021-09-02
> > > > > > > **Will be included in the revised manuscript**
> > > > > > >
> > > > > > > We will definitely include these results in the revised manuscript. Thanks for your support!

---

### Official Review · Reviewer_4dC4 · 2021-07-16

**Rating:** 7
**Confidence:** 3

**Summary:**

The authors address the challenge of training a single deep neural network, using a single loss function, which captures the specialized neural responses in both the ventral (“what”) and dorsal (“where”) pathways of mouse visual cortex. The authors demonstrate that a deep neural network comprised of two parallel convolutional hierarchies, trained with a single self-supervised predictive objective, captures neural responses in the ventral and dorsal streams better than a neural network comprised of a single convolutional hierarchy (which captures only the ventral stream), and better than either architecture trained with a supervised objective. Furthermore, by evaluating their two-pathway model on “what” and “where” downstream tasks —object categorization and motion detection— the authors show that their model has learned functionally specialized representations in each of its two pathways.

**Limitations And Societal Impact:**

The authors more than adequately address the limitations and potential societal impacts of their work, and suggest a few interesting directions for future work.

**Main Review:**

That the architectural choice to split a neural network into two parallel pathways encourages functionally specialized representations, even when both pathways are trained on the same objective, is on its own a remarkable and intriguing observation (although the authors do not appear to be the first to make this observation; see [1] for examples where branched architectures e.g. Alexnet exhibit specialization in their branches e.g. Gabor filters in one branch vs color detectors in another). This observation could inspire future works on the theory of deep learning, help to understand how representations are learned in neural networks, and aid in model interpretability. And it would be a resounding success if a simple anatomical constraint like this could boost the performance on specific tasks or encourage representations which better fit the brain (the authors demonstrate the latter under their representational similarity metric RSM).

However, I have one lingering question after reading the paper: has the 2-pathway architectural choice encouraged *new* functional specialization, or has it simply isolated functional specialization that already existed in the 1-pathway network? I would have liked to see a couple more analyses to investigate this possibility. (1) I would have liked to see the performance of the 1-pathway network on the two downstream tasks. Does the 1-pathway network already contain the specialization in its representations necessary to perform both downstream tasks successfully? Or does the 2-pathway constraint actually encourage new specialization that allows it to perform better on the downstream tasks? (2) I would have liked to see at least one other metric for comparing representations in the trained model to those in the brain. While the fact that the best layer in the 1-pathway network barely outperforms raw pixels at capturing the dorsal stream under the RSM is pretty convincing, naively I can imagine a scenario where the 1-pathway network contains both ventral-like and dorsal-like units, but the variance due to the ventral-like units swamps the variance due to the dorsal-like units, and hence the ventral-like features dominate the RSM as computed by Pearson correlation (whereas the 2-pathway network overcomes this by sorting the units into distinct subpopulations). Perhaps the authors could try another metric --such as regressing the target neural responses with a small number of source units, or under an L1 penalty, or CCA or something similar-- to determine whether dorsal-like features are present in the 1-pathway network (all analyses that have been used to compare models and brains in the past). (3) Finally, is there a clustering that could be performed, either on neural activity or connectivity, by which units in the 1-pathway network could be sorted into two functionally specialized subsets -- one which captures ventral-like features, and one which captures dorsal-like features?

The paper includes several other interesting insights, including recapitulating (to some extent) the anatomical hierarchy of mouse visual cortex, and indicating an apparent tradeoff between ventral-like specialization and dorsal-like specialization. Overall I think this work is a very interesting contribution, the analyses are well-performed, and the manuscript is clearly written.

[1] Voss, et al., "Branch Specialization", Distill, 2021.

**Time Spent Reviewing:**

8

---

> ### Author Response · Authors · 2021-08-10
> **Response to Reviewer 4dC4**
>
> We thank the reviewer for being supportive of our work and for the thoughtful and constructive comments. We will do our best to address the comments, and revise the paper accordingly. Please see our detailed responses to the comments below, along with our proposed changes to the manuscript:
>
> **“Has the 2-pathway architectural choice encouraged new functional specialization, or has it simply isolated functional specialization that already existed in the 1-pathway network?”**
>
> This is an excellent question. As a short answer, we have conducted some additional analyses of the models (see more details below) which suggest that the 2-pathway architecture (ResNet-2p) isolates and enhances functional specializations that already existed to some extent in ResNet-1p, in-line with the reviewer’s intuitions. As we explained in the paper, and showed in supplementary materials (section D), there seemed to be a trade-off or competition between dorsal-like and ventral-like representations throughout learning. The ventral-like representations dominated the competition by a significant margin in ResNet-1p (perhaps because they were more informative for the prediction task on this specific dataset), but it does not imply that there were no dorsal-like representations present in ResNet-1p. Adding a second pathway enabled the network to learn both types of representations more independently.
>
> Also the representational similarity analysis between ResNet-1p and ResNet-2p (Figure S3) showed that even though the one pathway of ResNet-1p was more similar to the ventral-like pathway of ResNet-2p (blue pathway), the dissimilarity between ResNet-1p and the dorsal-like pathway of ResNet-2p (red pathway) was not as large as the dissimilarity between the two pathways of ResNet-2p (Figure S2 - also compare the blue color areas in the similarity matrices in Figures S2 and S3). This observation suggests that, despite the dominance of ventral-like representations, there was a residue of dorsal-like representations in ResNet-1p.
>
> As to the additional analyses that we have run, we have conducted two new analyses of the model. In the first analysis, per the reviewer’s request, we have  examined the results for the downstream tasks with ResNet-1p. Here is a summary of the results: random-dots: [ResNet-2p (dorsal) = 74%, ResNet-2p (ventral) = 27%, ResNet-1p = 58%], CIFAR-10: [ResNet-2p (dorsal) = 33%, ResNet-2p (ventral) = 40%, ResNet-1p = 43%]. As can be seen, the one pathway architecture does better at the random dots task than the ventral-like pathway, but worse than the dorsal-like pathway, of the two pathway network. This suggests that ResNet-1p developed some dorsal-like functionality, but not as much as the dorsal-like pathway in the ResNet-2p model. We will include this data in the revised manuscript.
>
> It is also important to note, though, that the comparison on downstream tasks between ResNet-1p and each pathway of ResNet-2p is not a completely fair comparison, given that each pathway of ResNet-2p has half the number of output channels as ResNet-1p (64 channels for each pathway of ResNet-2p vs 128 channels for ResNet-1p). Thus, it is not an apples-to-apples comparison. Even though the difference between the accuracies of ResNet-1p and each pathway of ResNet-2p could be partly due to the difference in their output channel counts, we can say that in ResNet-1p both ventral- and dorsal-like specializations existed, and they were isolated and enhanced (especially for motion discrimination) in ResNet-2p. We will be sure to also highlight this in the revisions.
>
> The second additional analysis that we have conducted was to address the reviewer’s questions regarding the use of other metrics and clustering responses. Specifically, we fit two linear regression models to predict the response of every channel of ResNet-1p based on the responses of the neurons in (1) the most dorsal area (VISam) and (2) the most ventral area (VISlm). Then, we measured if the dorsal or ventral area is a better predictor for each channel. Throughout ResNet-1p, the dorsal area was a better predictor for a minority of channels (~30%), while the ventral area was a better predictor for the rest of the channels (and thus a majority). The same analysis for each pathway of ResNet-2p showed that the dorsal area of mouse visual cortex was a better predictor for 21% vs 64% of channels in the ventral- and the dorsal-like pathways, respectively.  This analysis, consistent with the other analysis outlined above, showed that there was a small subset of channels in ResNet-1p that were dorsal-like. In ResNet-2p, the ventral-like pathway contained even fewer dorsal-like channels, and the majority of the dorsal-like specializations were concentrated in the dorsal-like pathway. We will include these analyses in the revised manuscript.
>
> All in all, our results and additional analyses show that the dorsal-like representations do exist in ResNet-1p as well, but as shown in the paper and the analysis described above, the ventral-like representations are more dominant in the one-pathway model.

---

> > ### Comment · Reviewer_4dC4 · 2021-09-02
> > **thanks for your response**
> >
> > Thank you to the authors for their thorough response, and for the numerous new analyses they have performed to address my concerns. These new results are interesting and have confirmed what I suspected -- I think the revised manuscript will be clearer and even stronger with them included. I am still happy to recommend acceptance.

---

### Official Review · Reviewer_qYdj · 2021-07-16

**Rating:** 8
**Confidence:** 4

**Summary:**

The paper uses contrastive predictive coding to train deep networks on UCF101 videos. It then uses representational similarity to compare their representations to those of mouse calcium-imaging data from the Allen Brain Observatory. The key result is that a two-stream network is a good match to mouse visual cortex data. One of the (structurally identical) streams develops a representation that is a good match to dorsal mouse areas, and the other matches ventral mouse areas. Deeper areas correspond better to areas higher in the mouse visual hierarchy. Furthermore, the ventral stream of the model better supports transfer to CIFAR-10, while the dorsal stream better supports discrimination of random-dot motion direction.

**Ethical Concerns:**

The work does not raise ethical concerns.

**Limitations And Societal Impact:**

The work has a minor societal impact. The impact and limitations were thoughtfully addressed in the last section of the paper.

**Main Review:**

Originality:
There have been a few papers recently that compared deep-network representations to Allen Brain Observatory data. A recent one (Nayebi, A., Kong, N. C., Zhuang, C., Gardner, J. L., Norcia, A. M., & Yamins, D. L. (2021). Unsupervised Models of Mouse Visual Cortex. bioRxiv.) also uses contrastive objectives and considers networks with multiple streams. This paper and that one agree that contrastive objectives work well for modelling mouse data, but there is surprisingly little overlap otherwise. This paper focuses mainly on the issue of dorsal/ventral stream differentiation with a single cost function, while Nayebi et al. didn’t specifically address that point (although their StreamNets may have exhibited this differentiation).

Quality:
The technical approach is sound and the analysis is well-informed and appropriately thorough. In a couple of cases the description of the results seems more categorical than the results themselves. For example, line 223 says “unlike ResNet-1p, which fails to model the dorsal areas”. But ResNet-1p isn’t a total failure at the dorsal stream. It’s max RSM for VISal is even a little higher than that of ResNet-2p.

Clarity:
The paper is clear and well organized. The authors shared their code in case any details were missed.

Significance:
It is not too surprising that a network with two streams would develop some differentiation (Krizhevsky et al.’s version of AlexNet did this). However, it is significant that an unsupervised two-stream model with no other prodding would develop representations that are convincingly dorsal and ventral. The paper also contributes substantially to the understanding of what may lead to mouse visual representations, specifically.

Minor:
The paper was very clean. I only have a few suggestions:
-	Remind the reader in the caption of Figure 3 that red is dorsal.
-	Switch red and blue in Figure S4, to match the convention in the main paper.
-	Give parameters of the Gabor control model
-	Use vector graphics for the figures where possible. The file size is 10x that of the other papers I’m reviewing, and it doesn't scroll smoothly.


**Time Spent Reviewing:**

4 hours

---

> ### Author Response · Authors · 2021-08-10
> **Response to Reviewer qYdj**
>
>
> We thank the reviewer for their encouraging support of our work. We will rephrase the parts highlighted by the reviewer and double-check the rest of the manuscript to make sure our phrasing of the results captures the data more accurately. Also, we promise to address the minor points that the reviewer mentioned in their comments, including the adjustments to the figures and stating all of the relevant parameters.

---

### Official Review · Reviewer_3Khp · 2021-07-16

**Rating:** 7
**Confidence:** 4

**Summary:**

Motivated by the literature showing that the primate visual system has two pathways ("ventral" and "dorsal") with distinct functional specializations, the authors try to build a model of a visual system that also has representationally and functionally distinct regions. They turn to calcium imaging data from mouse visual cortex, which allows them to compare learned representations of ANN to the neural representations found in distinct cortical areas of animals shown the same images. (There aren't publicly available datasets with simultaneous recordings from ventral and dorsal areas in primates, as far as I know.) The authors find that a two-pathway 3D ResNet model trained on the contrastive predictive coding (CPC) task and a dataset of natural videos has layers that overall explain more of the mouse calcium responses in more cortical areas than several baseline models, including the same architecture trained on the same dataset but with a supervised task. In addition, the two pathways in the self-supervised model transfer better to different tasks (object recognition and motion discrimination), which is reminiscent of some of the functional specialization between primate ventral and dorsal pathways.

**Ethical Concerns:**

No, I have no concerns.

**Limitations And Societal Impact:**

I think what the authors wrote is fine, though to me the "Limitations" section reads more like "Here's what we would have done if we had more time" rather than "Here are the questions that our experiments weren't set up to address and are beyond the scope of the work." I think most of the things they suggest in the Limitations section are good ideas, and don't seem prohibitively expensive to try! I don't think I can reasonably expect them to try all of them in the week-long rebuttal period, but including at least some of them in a revision would make the paper much stronger.

**Main Review:**

I like this work overall and think the authors provide enough evidence to support their general conclusion: that it's **possible** for representationally and functionally specialized layers, resembling different regions of visual cortex, to emerge in a single computational model with the right architecture, trained on the right task. I'm not sure about two of their main specific claims, though: (1) that the particular architecture/loss function they identify here is the best model of mouse visual cortex that they could have reasonably identified, and (2) that their results have much to do with the distinction between the primate ventral and dorsal pathways. I'll discuss both of these below, but want to say up front that I think they can be remediated with some more careful model comparisons/controls and a clearer elaboration of the authors' central hypotheses. I'll start with (2) since it's a "higher level" response:

_Does the authors' work say something about "dorsal" and "ventral" visual pathways?_

As noted in the authors' introduction, ANNs optimized for object recognition (and recently in a self-supervised manner) are the best current models of multiple cortical areas in the primate ventral visual pathway. While there are a smaller number of cases where people have compared task-optimzied ANN representations to the primate dorsal pathway, the overall approach seems sound: these models predict neural activity or fMRI responses better than prior computational models, with some evidence that different tasks better account for dorsal areas than ventral areas and vice versa [A]. It would be nice, then, if there were single ANN models that could account for neural representations across both pathways in a "physical way" -- i.e., distinct layers of the single ANN could be convincingly mapped to distinct areas of visual cortex, extending the results of numerous studies for the ventral pathway alone (and primary visual cortex.)

But there's another problem that seems just as important to tackle, and much closer to what the authors are actually doing here: explaining the neural activity of _rodent_ visual cortex using task-optimized ANNs. To date, success has been mixed: there's some indication that object-recognition optimized deep CNNs do no better than randomly initialized models at predicting mouse visual cortex responses [B], and the top-layer features of such models do not seem to account for rat neural representations as well as early- and mid-layer features [C]. (I'm surprised that at least the first of these papers isn't discussed in the Related Work section.) Since the task-optimized ANN theory of visual representations isn't meant to be primate-specific, it predicts that **some** combination of architecture, task, and dataset should yield good predictions of rodent visual cortex (assuming task optimization methods succeed.) It seems reasonable to explore tasks that are more ethologically relevant to mice than complex invariant object recognition, which may be a primate-specific ability.

All this is to say: I'm surprised by the framing of this work as mainly addressing the first of these problems, not the second. I don't know that there's good reason to think of the various mouse visual areas as analogous to primate ventral and dorsal pathways (or whether there's evolutionary homology); the couple of references the authors give here might try to justify this categorization, but it's not clearly explained in this paper. In looking at Figure 1, I wouldn't immediately conclude that there are two distinct types of visual area in mice; one might just as easily propose that they all belong to a single hierarchy, with AM near the top and P/LM near the bottom. So, independent of the actual results (on which I'll comment below), I'm worried that the main claim here isn't well aligned with what the paper shows; comparing to the work in [A], for instance, I'm still more inclined to think that primate ventral and dorsal pathways are better explained by adaptation for **different** tasks rather than a single task with a branched (and then merging) architecture.

If the authors were to present this work more as addressing the second problem above -- actually explaining multiple regions of rodent visual cortex with a single task-optimized model -- I think it could be a clearer contribution to the literature on explaining neural data with task-optimized ANNs. If the authors are very attached to the dorsal/ventral distinction in mice, though, I think they need to justify the connection between mouse vision and (far better studied) primate vision much better than they do here.

_How close is the 2-stream CPC model to the "correct" architecture/task for explaining mouse visual cortex?_

Beyond the concern about framing (which I don't think is semantic or trivial -- it makes a big difference for ideas about the evolution of vision in mammals!) I have some questions about the actual experiments that I hope the authors could address in a revision. I think, overall, their results are pretty solid -- and pretty consistent with concurrent work on modeling mouse visual cortex [D]. Arriving at similar conclusions (multiple pathways, non-supervised object recognition tasks) despite many differences in implementation is a reassuring sign.

1. In the authors' initial model comparison (Figure 2), an ImageNet trained model ("Object Categ") did better than the untrained model at explaining responses in several the visual areas. This is a different result and conclusion than the one in [B]. Could the authors comment on what might have led to this difference?

2. The set of baselines compared to the 1- and 2-stream CPC models is pretty limited -- as the authors themselves mention in the "Limitations" section. I think the paper would be much stronger if there were a broader exploration of architectures trained on the same loss function and/or a broader set of loss functions tested for the same architectures, with the dataset held constant. For instance, the 2-stream model does seem better than the 1-stream model in that it has a better match to the AM representation in its higher layers, but (as per the discussion above) it's not at all clear from anatomy or prior work that 2 is the "correct" number of pathways here. Why not try a 3-, 4-, etc.-stream model? Or a PredNet-like model that makes predictions in pixel space rather than latent space? I appreciate that there are resource contraints here, but I don't think we can draw very strong conclusions about which architectures and/or losses are really the best match for mouse visual cortex until there's somewhat broader sampling of model space. (Especially if the authors are attached to the dorsal-ventral distinction, it would be good to see that 2 is the "correct" number of pathways.)

3. The object categorization model, if it's a pretrained ImageNet model, was likely trained at a different image resolution than the CPC models (unless I missed this somewhere.) It's also not the closest comparison to the CPC models that the authors could make. What about a self-supervised 2-stream CNN trained on a static contrastive loss, on the same dataset and input resolution? If that model also failed to predict RSMs in the "dorsal" areas, it would make for a much stronger argument that dynamic prediction is a key part of the ethologically relevant task.

4. It's very interesting that the two streams learn such different representations (judging by Figures 3 and 5) without any asymmetry between the two pathways' architectures other than different initial weights (though it's reminiscent of the original AlexNet.) It would be nice to get a qualitative flavor of how they differ. Do they learn very different convolutional filters at early layers or later layers? Are they really selective for shape versus motion as the authors suggest (the "dorsal" pathway is only a little worse at the CIFAR-10 task.) Anything to better understand how the symmetry is "broken" would be interesting to see here. Is it possible to bias one pathway to "become" the more dorsal one by changing the architecture slightly?

5. The authors use CPC as a proxy task for learning representations, but also try to argue (correctly, I think) that predicting the future of a visual scene is an ethologically important task. But we don't actually see in the paper whether the models are any good at this prediction problem, or whether the 2-stream architecture is better (gets lower loss) than the 1-stream architecture. If prediction is important and this one task/loss is supposed to explain neural processes across multiple visual areas, I would expect the authors to think that the 2-stream architecture is actually better at prediction than the 1-stream architecture! Otherwise, it doesn't seem like the 2-stream + CPC task provides an explanation of the structure of mouse visual cortex in the way that the authors want it to. So I'm eager to see whether this is the case.

6. Likewise, how does the 1-stream architecture do on transferring to the two tasks in Figure 5?

[A] Dwivedi, Kshitij, et al. "Unveiling functions of the visual cortex using task-specific deep neural networks." bioRxiv (2021): 2020-11.

[B] Cadena, Santiago A., et al. "How well do deep neural networks trained on object recognition characterize the mouse visual system?." (2019).

[C] Vinken, Kasper, and Hans Op de Beeck. "Using deep neural networks to evaluate object vision tasks in rats." PLoS computational biology 17.3 (2021): e1008714.

[D] Nayebi, Aran, et al. "Unsupervised Models of Mouse Visual Cortex." bioRxiv (2021).

**Time Spent Reviewing:**

4

---

> ### Author Response · Authors · 2021-08-10
> **(cont) Response to Reviewer 3Khp**
>
> **6- A Qualitative flavor of features in the two pathways**
>
> For a qualitative understanding of the difference between the two pathways of our model, we have created optimal stimuli for each pathway (using gradient ascent on the input space to maximize the layer activations). Since they were dynamic visualizations, there was no straightforward way of including them in the paper, but we will include an external link to these visualizations in the supplementary materials.
>
> Here are some visualizations for the ventral-like and the dorsal-like pathways of ResNet-2p: https://files.catbox.moe/86818l.zip. Please unzip the compressed folder and open the html file in your browser.
>
> The stimuli are created by maximizing the activation of the layers with the highest similarity with VISlm (for the ventral-like pathway) and VISam (for the dorsal-like pathway). The optimal stimuli for the ventral-like area mostly contain static textures and patterns with some local moving components, while for the dorsal-like area, the optimal stimuli mostly contain distributed motion.
>
> **7- Predictive performance of CPC with ResNet-1p vs ResNet-2p**
>
> This is an excellent comment. The predictive performance of the two models were not significantly different (both ~90% top-3 accuracy). But, it is important to note that ResNet-2p had, in total, a lower number of parameters than ResNet-1p (ResNet-1p: 435k  vs ResNet-2p: 285k). The total number of channels was kept equal in the two architectures, but given that there was no interconnection between the two pathways of ResNet-2p, ResNet-1p has about 1.5 times more parameters in total than ResNet-2p. Therefore, considering the lower capacity of ResNet-2p and its similar predictive performance to ResNet-1p, we can conclude that the inductive bias of parallel architecture may have helped in predictive processing. In other words, with a smaller model, we can reach the same level of predictive performance if we use an architecture that has parallel pathways. We will note this in the revised manuscript.
>
> **8- One-stream architecture on downstream tasks**
>
> We will include the results of downstream tasks on ResNet-1p in the revised manuscript. Here is a summary of the results: random-dots: [ResNet-2p (dorsal) = 74%, ResNet-2p (ventral) = 27%, ResNet-1p = 58%], CIFAR-10: [ResNet-2p (dorsal) = 33%, ResNet-2p (ventral) = 40%, ResNet-1p = 43%].
>
> It is important to note though that the comparison between ResNet-1p and each pathway of ResNet-2p is not a fair comparison, given that each pathway of ResNet-2p has half the number of output channels as ResNet-1p (64 channels for each pathway of ResNet-2p vs 128 channels for ResNet-1p). Thus, it is not quite an apples-to-apples comparison. Even though the difference between the accuracies of ResNet-1p and each pathway of ResNet-2p is partly due to the difference in their output channel counts, we can say that in ResNet-1p both ventral- and dorsal-like specializations existed, and they were isolated and enhanced (especially for motion discrimination) in each pathway of ResNet-2p. We will include these results along with these explanations in the revised manuscript.

---

> ### Author Response · Authors · 2021-08-10
> **Response to Reviewer 3Khp**
>
> We thank the reviewer for being supportive of our work and for the thoughtful and constructive comments. We will do our best to address their comments, and revise the paper accordingly. We note, though, that at NeurIPS 2021 we cannot submit a revised paper at this time, so we will merely highlight potential revisions and additions that we promise to make for a camera ready version.
>
> Below are our detailed responses to the reviewer’s comments:
>
> **1- Regarding the ventral/dorsal streams in primates vs rodents**
>
> This is a critical issue, and we appreciate the reviewer bringing it out. We will include additional clarifications on this topic in the revised manuscript, and be careful to highlight the point that the homology between rodent and primate visual cortex may not be clear. Nonetheless, here is an explanation of why we believe keeping the dorsal/ventral theme in the paper is appropriate, and at the same time, can also serve the purpose of forming a better understanding of the mouse visual system:
>
> We agree with the reviewer that the dorsal/ventral mapping of rodents’ visual system is not as clear and well studied as in primates. We also agree that our study could be framed as a model of modular organization of the mouse visual system without referring to the dorsal/ventral streams. Importantly, though, we believe that our results do speak to functional specialization between brain regions that are primarily concerned with static objects versus movement. As observed in several papers (Macdonald and Mascagni 1996, Neuroscience; Marshal et al. 2011, Neuron; Wang et al. 2011, Journal of neuroscience; Wang et al. 2012, Journal of Neuroscience; Laramee and Boire 2015, Frontiers in neural circuits), rodents also possess such specialized pathways. The reviewer is correct that these pathways in rodents may not be perfectly homologous to primate ventral and dorsal pathways. But nonetheless, by studying the mouse visual system, which possesses these functional specializations, we believe that our work also speaks to the specialization between static objects and movement seen in primate ventral and dorsal pathways, respectively. We would also like to note that a functional homology between the parallel pathways in mouse visual cortex and the dorsal/ventral pathways in primates’ visual system is a theory that several previous studies had suggested (e.g. Glickfeld et al. 2014, Current opinion in neurobiology; Saleem 2020, Current opinion in neurobiology) and which has some anatomical and functional support (Marshal et al. 2011, Neuron; Wang et al. 2011, Journal of neuroscience; Sit and Goard 2020, Nature communications). Thus, we would like to emphasize that we are not the first researchers to propose some homology between the distinct pathways in rodent visual cortex and primate visual cortex. We would also add that since we tested the two pathways of our model on downstream tasks that were informed by our knowledge of the ventral and dorsal pathways in primates, our claim that the functional specialization in our two-pathway ANN has some relevance to the ventral and dorsal pathways is supported by the data. But, we take the reviewer’s point, and promise to revise the paper with these issues more clearly delineated so as to allow the reader to understand the complexity and potential pitfalls of using data from mouse visual cortex to make broader interpretations about ventral and dorsal pathways. As well, we promise to highlight, as the reviewer suggests, the fact that our model clearly does speak to the functional specialization in rodent cortex, even if it turns out that the previous theories positing a homology between rodent and primate visual cortex are incorrect.
>
>
> **2- “I'm still more inclined to think that primate ventral and dorsal pathways are better explained by adaptation for different tasks rather than a single task with a branched (and then merging) architecture.”**
>
> This could very well be the case, and we believe that our study does not dismiss this possibility. What we essentially showed in this paper is that a predictive loss function and an architecture with parallel pathways is sufficient for modeling the specialization of the visual system. We cannot claim though that this combination of loss and architecture is necessary, though. As mentioned by the reviewer, an alternative approach would be to have a separate loss function for each pathway that models their downstream functional requirements. In our estimation, it will likely be a combination of a single predictive loss and some individual downstream losses that would best explain the functional properties of the mammalian visual system, and it may be that different species require different combinations of losses. A thorough comparison needs to be done between all these options to figure out which model best explains the visual system of each species. This is definitely a critical future step which we are currently working on. Unfortunately, giving a solid and rigorous answer to this question requires more time than provided in the NeurIPS revision period, and we believe that it is a bit outside the scope of this paper. However, given the clear possibility that the reviewer is correct, we will be sure to highlight in the revised paper the point that our model is showing only sufficiency for such functional specialization, and not necessity. Thus, it is possible, likely even, that other models using distinct tasks for the two pathways could also fit the ventral/dorsal specializations.
>
> **3- The results on ImageNet-trained model is different from those reported in [B]**
>
> This is a good point, thanks for bringing it up. It is difficult to precisely identify the reasons behind the difference between our results and those of [B]. Some initial, clear differences are the architecture of the network used and the datasets used for the RSA. On that second item, we suspect that a key difference is the visual stimuli used in our study and those used in [B]. In our study natural videos were used as visual stimuli, while static natural images were used in [B]. As shown by de Vries et al (2020) (reference [8] in the paper), dynamic natural stimuli are the optimal stimuli for eliciting responses from different areas of mouse visual cortex. Static images are likely to not match the temporal receptive fields and the preferred temporal frequencies of neurons in mouse cortex very well, and therefore, do not elicit sufficiently strong responses that can then distinguish between the representations of the ImageNet-trained and randomly initialized ANNs. We will cite [B] and note these differences in the revised manuscript.
>
> **4- What about architectures with more than two parallel pathways?**
>
> We also investigated models with more than two parallel pathways. The architecture with two parallel pathways showed the highest average similarity with all the areas. We will include these comparisons in the supplementary materials of the revised manuscript, and will mention this analysis briefly in the main text.
>
> However, these results do not suggest that the functional structure of the mouse visual system should be understood with only two parallel pathways. The anatomy of the rodents’ visual system may also suggest otherwise. The optimality of two parallel pathways in our results could be a limitation of the datasets that we used in this study (both training videos and the neuronal dataset). Backbone architectures with more than two parallel pathways could potentially generate more similar representations to mouse visual cortex if we used training videos that better capture the natural visual experience of mice than the UCF-101 dataset. The limitation of the neuronal dataset is another factor that could affect these results. The (dis)similarities of different pathway representations are highly affected by the task and visual experience the animal had been engaged in during the experiments. There could be more parallel modes of visual processing in mouse visual cortex that would be activated if, for example, mice were engaged in a more active task that required whisking behavior, reaching, or navigating. The richer the task conditions, the higher the chance of finding the correct model of the animal’s visual system. Even in primates, the dorsal/ventral dual pathways is most likely a simplification of the actual anatomical and functional architecture of the visual system, but it is a parsimonious theory that best explains the available data. As noted, we will include the other multi-pathway results in the revised supplemental, and add some discussion on these issues.
>
> **5- More baseline models would be better, including a self-supervised learning model with static images**
>
> We agree with the reviewer that more baseline models would help the paper. First, we will include comparisons with the SlowFast network (Ref 12 in the paper) and other 3D convolutional neural networks trained on action recognition in the supplementary materials of the revised manuscript, based on requests from other reviewers. Second, per this reviewer’s request, we will include a self-supervised model trained on static images in the revised manuscript. We agree that this is an important control, as currently we have a control where we keep the dataset the same but change the loss, yet we don’t have a control where we switch the dataset but maintain a similar loss.  It should be noted that self-supervised training on static images could not produce the spatiotemporal selectivity we observed in our dorsal-like pathway, but it will serve as an excellent comparison for the ventral-like pathway. It may even provide as good or better a fit to regions like LM. The models are currently being trained, and we will update the reviewers when the results are available and include these new baselines in the revised manuscript.

---

> ### Author Response · Authors · 2021-09-01
> **Response to Reviewer 3Khp: Comparisons with a SSL model trained on static images**
>
> To address the reviewer's comment regarding self-supervised learning with static images, we used SimCLR self-supervised learning to train ReNets with one or two parallel pathways on STL10 dataset (image sizes 64 by 64). In short, for ventral areas, SimCLR performed around the same level as the ImageNet-trained models (Object Categ in Figure 2c). However, unlike CPC, neither one- nor two-pathway architectures trained with SimCLR could explain the dorsal representations. CPC, in general, showed higher representational similarity with all the areas of mouse visual cortex, which supports the hypothesis that the predictive component of the CPC loss function and the spatiotemporal dynamics of video data aid in modeling mouse visual cortex more accurately. The details of these results will be included in the revised manuscript. Below are the maximum representational similarity values (similar to Figure 2c) for the ResNet-2p architectures trained with SimCLR and CPC:
>
> VISlm:  (**SimCLR**): 0.576 (0.0252) &nbsp; &nbsp; (**CPC**): 0.628 (0.0125)
>
> VISp:    (**SimCLR**): 0.676 (0.0170) &nbsp; &nbsp; (**CPC**): 0.722 (0.016)
>
> VISpm:  (**SimCLR**): 0.520 (0.019) &nbsp; &nbsp; (**CPC**): 0.582 (0.011)
>
> VISal:    (**SimCLR**): 0.6621 (0.020) &nbsp; &nbsp; (**CPC**): 0.691 (0.016)
>
> VISam:  (**SimCLR**): 0.590 (0.005) &nbsp; &nbsp; (**CPC**):0.694 (0.010)

---

### Author Response · Authors · 2021-08-10
**General Response**

We would like to thank all of the reviewers for their thoughtful comments on our paper. We were pleased that the scores indicated that the reviewers generally felt the paper was a good contribution to the literature. The reviewers made some very insightful comments and recommendations, and we have attempted to address all of them here with discussion, promises for revision to the writing, and some new data. There were a couple of suggestions to train additional models, but we did not have enough time in this first review period for that. In such cases, we have identified our plans for revisions and described them to the reviewers. We believe that with our planned revisions the paper will be improved, and we thank everyone for their help with this.

---

### Decision · Program_Chairs · 2021-09-27

**Decision:**

Accept (Spotlight)

**Comment:**

This is a clear accept.  All the reviewers were clearly positive, and I largely agree with their judgements.   I think the reviews speak for themselves, and I don't have much to add here.